# Inferring surface energy fluxes using drone data assimilation in large eddy simulations

Norbert Pirk[1], Kristoffer Aalstad[1], Sebastian Westermann[1], Astrid Vatne[1], Alouette van Hove[1], Lena Merete Tallaksen[1], Massimo Cassiani[2], and Gabriel Katul[3]

[1]Department of Geosciences, University of Oslo, Sem Sælands vei 1, 0371 Oslo, Norway
[2]NILU - Norwegian Institute for Air Research, Instituttveien 18, 2007 Kjeller, Norway
[3]Department of Civil and Environmental Engineering, Duke University, 121 Hudson Hall, Durham, NC, 27708, USA

**Correspondence:** Norbert Pirk (norbert.pirk@geo.uio.no)

**Abstract**

Spatially representative estimates of surface energy exchange from field measurements are required for improving and validating Earth system models as well as satellite remote sensing algorithms. The scarcity of flux measurements can limit understanding of ecohydrological responses to climate warming, especially in remote regions with limited infrastructure. Direct

field measurements often apply the eddy covariance method on stationary towers, but recently drone-based measurements of temperature, humidity, and wind speed have been suggested as a viable alternative to quantify the turbulent fluxes of sensible ($H$) and latent heat ($LE$). A data assimilation framework to infer uncertainty-aware surface flux estimates from sparse and noisy drone-based observations is developed and tested using a turbulence-resolving large eddy simulation (LES) as a forward model to connect surface fluxes to drone observations. The proposed framework explicitly represents the sequential

collection of drone data, accounts for sensor noise, includes uncertainty in boundary and initial conditions, and jointly estimates the posterior distribution of a multivariate parameter space. Assuming typical flight times and observational errors of light-weight, multi-rotor drone systems, we first evaluate the information gain and performance of different ensemble-based data assimilation schemes in experiments with synthetically generated observations. It is shown that an iterative ensemble smoother outperforms both the non-iterative ensemble smoother and the particle batch smoother in the given problem, yield-

ing well-calibrated posterior uncertainty with continuous ranked probability scores of $12 \ \mathrm{W \ m^{-2}}$ for both $H$ and $LE$ with standard deviations of $37 \ \mathrm{W \ m^{-2}}$ ($H$) and $46 \ \mathrm{W \ m^{-2}}$ ($LE$) for a 12 min vertical step profile by a single drone. Increasing flight times, using observations from multiple drones, and further narrowing the prior distributions of the initial conditions, are viable to reducing the posterior spread. Sampling strategies prioritizing space-time exploration without temporal averaging, instead of hovering at fixed locations while averaging, enhance the non-linearities in the forward model and can lead to

biased flux results with ensemble-based assimilation schemes. In a set of 18 real-world field experiments at two wetland sites in Norway, drone data assimilation estimates agree with independent eddy covariance estimates, with root-mean-square error values of $37 \ \mathrm{W \ m^{-2}}$ ($H$), $52 \ \mathrm{W \ m^{-2}}$ ($LE$), and $58 \ \mathrm{W \ m^{-2}}$ ($H + LE$), and correlation coefficients of $0.90$ ($H$), $0.40$ ($LE$), and $0.83$ ($H + LE$). While this comparison uses the simplifying assumptions of flux homogeneity, stationarity, and flat terrain, it is

emphasized that the drone data assimilation framework is not confined to these assumptions and can thus readily be extended
to more complex cases and other scalar fluxes, such as for trace gases in future studies.

# 1 Introduction

The significance of sensible ($H$) and latent ($LE$) heat fluxes between an underlying surface and the atmosphere aloft is not in dispute, given the plethora of problems spanning atmospheric, oceanographic, cryospheric, soil, and vegetation dynamics, in which these turbulent exchange processes arise. Direct measurements of these surface fluxes enable robust methods to evaluate
and tune parametrizations used in climate models, and to develop algorithms for indirect flux retrieval using satellite remote sensing. Traditionally, flux measurements are collected on meteorological towers using the so-called eddy covariance (EC) technique (or other micro-meteorological approaches such as the Bowen ratio method). While these stationary tower measurements are often considered the best available technique for surface flux estimation, they are known to have limited spatial representativeness (Chu et al., 2021). Moreover, the link between measured turbulent heat flux at the tower and sources or sinks
at the surface becomes problematic when these sources and sinks are spatially variable (Bou-Zeid et al., 2020). An indirect manifestation of this problem is a failure to close the surface energy balance with underestimates in excess of 20% (on average) being reported across the sites of the FluxNet network (Stoy et al., 2013). This problem is by no means confined to surface heterogeneity. A number of studies (Steinfeld et al., 2007; De Roo et al., 2018) showed that organized eddies can bias flux estimates even in homogeneous environments, which explains part of the typically observed lack of closure of the measured
surface energy balance. One approach to ameliorate these issues is to include spatially distributed measurements, which frames the scope of the work herein. Airborne measurements from aircrafts have a long tradition in atmospheric sciences and are used to complement flux towers (Desjardins et al., 1989; Mahrt, 1998). Over the past decade, developments in miniaturized sensors and small unoccupied aircraft systems (hereafter referred to as drones) have been opening possibilities for studies of land-atmosphere interactions in ways not attempted before.
Drones measuring air temperature, humidity, and wind speed are a promising tool for spatially distributed measurements in meteorological studies (Lee et al., 2018; Barbieri et al., 2019). Fixed-wing drones are typically equipped with air speed sensors that allow for wind speed estimation (Elston et al., 2015). Multi-rotor drones introduce some distortions to the turbulence field around them but they can still estimate horizontal wind speed based on their altitude derived from inertial measurement unit (IMU) data (Neumann and Bartholmai, 2015; Palomaki et al., 2017). Drones can also 'hover' in place at a point of interest
or can sample along a pre-programmed flight path making them a sort of intermediate between tethered balloon soundings and helicopter platform. Drone data have already been used for flux estimation in several studies (Bonin et al., 2013; Hoffmann et al., 2016; Kim and Kwon, 2019; Båserud et al., 2020), typically using Monin-Obukov similarity theory (MOST) (Monin and Obukhov, 1954; Foken, 2006) with flux-profile (Högström, 1988) or flux-variance (Katul and Hsieh, 1999) relationships as well as vertically-integrated heating/drying rates to infer surface fluxes. Due to the stochastic nature of turbulent trans-
port, measurements are usually aggregated over longer time periods or large spatial distances where the statistical variability becomes predictable by micrometeorological theories (though ergodicity is a priori assumed in this case). Nonetheless, given

the limited flight time of drones, new trade-offs in the spatio-temporal sampling strategy could be developed to optimize flux estimates. Recently, multi-platform systems or drone swarms, carrying a mobile sensor network, have been shown to have capabilities for estimating emissions from gas point sources (Hutchinson et al., 2017; Ristic et al., 2020). While the potential for drone-based flux measurements as a relatively low-cost and mobile complement to EC is promising, there are many open questions regarding the uncertainties of the resulting flux estimates, the optimal flight strategy, the required turbulent transport model, and the data-model fusion algorithms.

Returning to the issue of spatial variability and scales, the surface layer of the atmosphere constitutes a non-linear system where variability exists across all scales (Wyngaard, 2010). To explicitly represent intermittent and inhomogeneous turbulent transport as well as coherent structures requires high-resolution models that are computationally much more expensive than the flux-related expressions encoded in MOST. In particular, turbulence-resolving large eddy simulations (LES) are widely accepted tools to simulate boundary layer dynamics, as they explicitly resolve the energy-containing range of large eddies while they parametrize the effect of sub-grid scales on the resolved scales (see e.g., Stull, 1988). While MOST describes planar-homogeneous and stationary turbulence statistics in the absence of subsidence, LES allows for the analysis of turbulence time series at high temporal resolution, so as to realistically represent turbulence statistics collected at time scales of seconds to minutes. Some studies have already paved the way (Sühring et al., 2019) by performing idealized LES studies with known initial and boundary conditions, and with virtual airborne measurements to show the feasibility of airborne flux estimation techniques, even above heterogeneous surfaces (but disregarding sensor noise). This indicates that drone observations can be combined with LES to estimate surface fluxes. That is, the LES may be viewed as a mathematical operator that takes surface boundary conditions and key large-scale meteorological forcing and provides statistics such as turbulent fluxes and meteorological states at all points in the domain of interest over a period of time. These statistics can then be compared with 'noisy' data obtained from drones. Surface fluxes that optimally match the noisy measurements can then be inferred.

This view implies that a mathematically optimal technique for consistent data-model fusion can be formulated as a kind of Bayesian inference problem (MacKay, 2003; Jaynes, 2003; Särkkä, 2013; Gelman et al., 2013), which is typically referred to as data assimilation (DA) or inverse modeling in the geosciences (Carrassi et al., 2018; Evensen et al., 2022b). Herein, we adopt a broad Bayesian definition of the field of DA in line with Evensen et al. (2022a). In addition to the conventional DA problem of state estimation, this definition also encompasses the problem of parameter estimation. The latter is often referred to as an inverse problem (Stuart, 2010) rather than a DA problem. Since the flux estimation problem at hand is precisely such a parameter estimation or inverse problem we are also leaning on developments in this field (Iglesias et al., 2013; Schillings and Stuart, 2017). In this study, we do not make any distinction between DA and inversion and take a unifying approach through the lens of Bayesian inference following Särkkä (2013). Such a unified view is especially helpful as the methods used herein can be applied in a hierarchical framework that jointly solve both state and parameter estimation problems (Katzfuss et al., 2020).

On regional scales, DA with mixed-layer models have been used to estimate surface energy fluxes from surface temperature measurements provided by satellite remote sensing (Caparrini et al., 2004; Xu et al., 2018) or radiosonde profiles of potential temperature and specific humidity (Tajfar et al., 2020). On much larger scales, Bayesian flux inversions are also a common tool

in atmospheric inverse modeling to assess regional emissions of $CO_2$ (Tans et al., 1989) and methane (Thompson et al., 2018). Due to computational limitations, studies using DA with LES models have only recently become possible (Lunderman et al., 2020), and an evaluation with drone observations together with independent flux measurements remains lagging (or lacking all together).

In many practical applications, one typically omits stochastic terms in the model and assumes it to be a perfect representation of reality (so-called strong-constraint data assimilation) (Evensen, 2019). Even so, different DA techniques will excel depending on model complexity and the number of parameters in the problem. Variational DA combines the model and the data through the optimization of a cost function, but requires taking derivatives of the forward model with respect to its parameters (Bannister, 2017), which is difficult or impossible for most LES codes. Particle-based methods (van Leeuwen et al., 2019), such as the particle batch smoother scheme (Margulis et al., 2015), are conceptually well suited for drone data assimilation given their limited assumptions, but they are known to suffer from degeneracy for problems in higher dimensional parameter spaces (Snyder et al., 2008). Ensemble Kalman-based methods, such as the ensemble smoother (van Leeuwen and Evensen, 1996) and its iterative variants (Emerick and Reynolds, 2013), on the other hand, have been shown to overcome some of these limitations for very large parameter spaces. However, these approaches invoke Gaussian linear assumptions at the analysis phase when data and models are combined. These assumptions can be problematic given that Gaussian random variables do not respect physical bounds and many forward models in the geosciences are non-linear. This issue motivated studies that sought to describe how the iterative ensemble Kalman smoother can be used to improve urban air pollution estimation by assimilating both mean wind and gas concentrations with a Reynolds-Averaged Navier-Stokes (RANS) model (Defforge et al., 2021). Considering the potential and limitations of the different DA schemes, one may hypothesize that either Ensemble Kalman-based or particle-based approaches (or a combination thereof) could be ideal for drone data assimilation in LES.

The aim of the present study is to first perform observing system simulation experiments (Masutani et al., 2010) to evaluate which DA scheme is most suited for the problem of flux estimation from drone observations, and to demonstrate what flux results can be expected from typical light-weight drone systems. We then apply the drone data assimilation technique to real-world measurements from drones and compare its results to concurrent eddy covariance flux estimates to demonstrate the feasibility of the method. To be clear, given the differences in footprint and underlying assumptions, we do not argue that this comparison offers a validation per se – only a plausibility check of the estimated order of magnitude of fluxes and their relative variability.

## 2 Materials and methods

### 2.1 Data assimilation framework

The aim here is to infer surface fluxes of sensible and latent heat using sparse and uncertain drone measurements of meteorological variables in the atmospheric boundary layer. Solving this inverse problem requires a forward (or data generating) model that maps the parameters, namely the surface fluxes of interest and other uncertain boundary conditions, to the drone

observations through

$$\mathbf{y} = \mathcal{G}(\mathbf{x}) + \boldsymbol{\epsilon}, \tag{1}$$

where $\mathbf{y} \in \mathbb{R}^d$ is the observation vector, $\mathcal{G}(\cdot)$ is the forward model, $\mathbf{x} \in \mathbb{R}^m$ is the target parameter vector, and $\boldsymbol{\epsilon} \in \mathbb{R}^d$ is the observation error. In practice, $\mathcal{G}(\cdot)$ is a composition of multiple operations (c.f. Evensen et al., 2022b)

$$\mathcal{G}(\mathbf{x}) = \mathcal{H}(\mathcal{M}(\mathcal{T}(\mathbf{x}))) . \tag{2}$$

The inner operation, $\mathcal{T}(\cdot)$, is a transformation step that maps the parameters from an unbounded space to a bounded physical space. This step helps satisfy the Gaussian assumption of the ensemble Kalman methods while avoiding unphysical values (Section 2.1.2), although it adds an extra layer of non-linearity to the forward model. The subsequent middle operation, $\mathcal{M}(\cdot)$, is the dynamical model used to simulate the state of the boundary layer given the boundary conditions specified by the parameters. The outer operation, $\mathcal{H}(\cdot)$, is the observation operator that maps the states of the model to the corresponding predicted observations by extracting the flight paths of drones and (when necessary) performing temporal aggregation (see Section 2.1.3). By employing a turbulence-resolving LES as opposed to a RANS model for the dynamics $\mathcal{M}(\cdot)$ in our forward model $\mathcal{G}(\cdot)$, we are able to generate the surface flux to drone observation mapping since the LES is run at an appropriate level of spatio-temporal detail.

Even in the absence of observation error, the inversion of Equation (1) will typically be an ill-posed problem in the sense that a solution for the parameters $\mathbf{x}$ may not exist or be unique (Stuart, 2010). As such, it is more instructive to abandon the quest for a single optimal solution, which does not necessarily exist in a well-defined way, and rather approach this problem in a probabilistic manner using Bayesian inference (Jaynes, 2003; MacKay, 2003; Särkkä, 2013). We do this following a classical Bayesian approach in geosciences known as data assimilation (DA) reviewed elsewhere (Wikle and Berliner, 2007; Evensen et al., 2022b), where we use a prior distribution $p(\mathbf{x})$ to represent our knowledge concerning possible values for the model parameters $\mathbf{x}$ before taking the observed drone data $\mathbf{y}$ into account. We combine this with a second distribution, the likelihood $p(\mathbf{y}|\mathbf{x})$, which describes the probability of generating the data for a given set of parameters of the LES model. To help construct this likelihood, conventional DA assumptions are followed (e.g. Carrassi et al., 2018) by using an additive Gaussian observation error $\boldsymbol{\epsilon} \sim \mathrm{N}(\mathbf{0}, \mathbf{R})$ with zero mean and observation error covariance matrix $\mathbf{R}$. Bayes' theorem then yields a posterior distribution of the parameters $p(\mathbf{x}|\mathbf{y})$ by taking the product of prior and likelihood, i.e.

$$p(\mathbf{x}|\mathbf{y}) = \frac{p(\mathbf{y}|\mathbf{x})p(\mathbf{x})}{p(\mathbf{y})}, \tag{3}$$

which represents our knowledge of the parameters and their uncertainties in view of our uncertain prior knowledge as well as the data and their assumed error distribution. The so-called model evidence $p(\mathbf{y})$ in the denominator of Equation (3) simply plays the role of a normalizing constant in this context. To solve this probabilistic inverse problem in practice, various derivative-free ensemble-based DA schemes can be used to estimate the posterior numerically by adopting particle and/or Gaussian approximations.

This problem formulation is implicitly conditioned on the strong constraint (see Evensen et al., 2022b) that the forward model $\mathcal{G}$ is a perfect representation of reality. As George Box humorously notes – even though all models are wrong, what

matters is the extent to which they are useful (Box, 1976). From this perspective, synthetic experiments (described below) are useful because the models are perfect by construction and thus useful for testing and comparing the DA algorithms. In the real experiments, where we compare with independent EC data, some of the mismatch between the EC estimates and drone-based inferences will undoubtedly be due to the strong assumptions made in the respective approaches. Given the level of realism in LES, these structural model errors introduced when moving the algorithm application from synthetic to field data are likely dominated by simplifications of topography and spatio-temporal flux variability. The Bayesian approach to inference also offers a way to compare the relative usefulness of different models using the model evidence (MacKay, 2003), although this will not be pursued here.

### 2.1.1 LES model and parameters

The turbulence-resolving Parallelized Large-Eddy Simulation Model (PALM) (Raasch and Schröter, 2001; Maronga et al., 2015) version 6.0, is used as the forward model. PALM solves the filtered Navier-Stokes equations and the first law of thermodynamics with the Boussinesq approximation to explicitly resolve turbulent motions in the atmospheric boundary layer. The effect of sub-grid scale motions on the flow is parameterized using the kinetic energy scheme of Deardorff (1980) as the sub-grid model. It is widely used in the boundary layer community to simulate neutral, stable, and unstable boundary layers (Steinfeld et al., 2007; Couvreux et al., 2020) as well as scalar transport (Ardeshiri et al., 2020).

The number of grid points in the simulations is set to 256 by 256 longitudinally (along $x$) and laterally (along $y$), and 128 vertically (along $z$). The planar grid spacing is 10 m. Vertically, the grid spacing is 5 m between the surface and the height of 240 m, above which a grid stretching of 1.03 is applied. Thus, the modeling domain is 2560 m by 2560 m in the $x - y$ plane, and 1950 m vertically. The computational grid is chosen to be sufficiently fine to explicitly resolve small scale unorganized turbulence so that the sub-grid fluxes are small compared to resolved-scale fluxes, even relatively close to the surface. The size of the model domain is large enough to include the evolution of large scale organized structures that can form in convective boundary layers and to minimize the formation of superstructures that are larger than the domain. Cyclic lateral boundary conditions are applied. Between the surface and the first grid level, a constant flux layer with MOST (i.e. with stability correction) is assumed to connect the surface to the atmosphere. Following Sühring et al. (2019) each simulation starts with a constant potential temperature and specific humidity profile to a height of 800 m, above which a capping inversion with a vertical gradient of 1 K per 100 m for potential temperature and $-0.5$ g kg$^{-1}$ per 100 m for specific humidity is used. To facilitate comparison, we use the same simplifying assumptions as EC, namely homogeneity and stationarity of surface fluxes, and flat terrain.

Boundary and initial conditions for $H$, $LE$, aerodynamic roughness length ($z_0$), initial potential temperature ($\theta_{\text{init}}$), initial specific humidity ($q_{\text{init}}$), and geostrophic wind speed at the surface ($u_g$) are varied in the LES ensemble simulations according to prior distributions for each parameter (described below). Of these six parameters, the primary interest is in $H$ and $LE$ while the remaining four parameters can be regarded as 'nuisance' parameters (Bretthorst, 1988; Jaynes, 2003; Gelman et al., 2013). The nuisance parameters are still inferred from the data, but are then implicitly 'integrated out' as we primarily focus

on the marginal posterior distributions of $H$ and $LE$. Due to the planar-homogeneous surface, there is no need to use an extra parameter for the second lateral component of the geostrophic wind speed at the surface.

Each simulation starts with a spin-up period during which turbulence generation is triggered by adding artificial random perturbations until turbulence starts to develop freely. The time series of the maximum vertical wind velocity and the resolved-scale turbulence kinetic energy shown in Figure S1 (see Supplementary material) indicate that $4680$ s typically suffices to achieve stationary turbulence statistics in most simulations (corresponding to about $10$ eddy turnover times). Some ensemble members will represent parameter combinations that hardly allow for a turbulent flow regime, e.g. during strongly stable conditions with very negative sensible heat fluxes and low geostrophic wind speeds, and will therefore not develop stationary turbulence. Some of the prior parameter combinations might in reality also be physically improbable and would therefore yield extremely unlikely model predictions. Consequently, the inferred posterior probability will be low for such cases given that the drone data was generated under different regimes.

Figure 1 shows examples of ensemble members from an ensemble of LES simulations as cross sections of potential temperature after the spin-up period. Heating of the surface induces thermal convection in organized structures that works together with shear-driven (mechanical) turbulence to transport heat away from the surface and momentum towards the surface. On average, this boundary layer gradually warms up and humidifies over time, in a manner that can be considered quasi-stationary after spin-up. Spatial differences of about $1.0$ K can be seen in the surface layer in this simulation. At the top of the boundary layer, warm and dry air is mixed into the boundary layer (entrainment), while the capping inversion effectively limits turbulent mixing further up. The $x - y$ cross sections (Figure 1, right) show a few of the spatial structures that are typically included in the ensemble.

### 2.1.2 Prior distributions

The prior distributions for $H$ and $LE$ are set to be normal (i.e. Gaussian) distributions centered at $0$ with standard deviations of $150$ W m$^{-2}$ each. For $u_g$ and $z_0$, log-normal prior distributions were specified (to ensure strictly positive support) with means (of the underlying normal distribution) of $0.7$ and $-1.2$, and standard deviations of $0.7$ and $0.5$, respectively. The priors for $\theta_{\mathrm{init}}$ and $q_{\mathrm{init}}$ are set to be normal distributions with mean values that are determined separately for each experiment to account for the large differences in mean temperature and humidity between our experiments. For these variables only, we follow the empirical Bayesian approach to constructing priors (Murphy, 2022) and determine these mean values from the drone observations themselves based on the observed temperature and humidity range. For the synthetic experiments, we chose priors for $\theta_{\mathrm{init}}$ and $q_{\mathrm{init}}$ that include the true values, but are not centered on them (approximately $0.5$ standard deviations offset). This bias of the priors for these nuisance parameters makes subsequent inference more challenging and realistic. The standard deviation of the priors of $\theta_{\mathrm{init}}$ and $q_{\mathrm{init}}$ are $0.3$ K and $0.1$ g kg$^{-1}$, respectively. These values relate to the observation errors described below. To test the sensitivity to the uncertainty in initial conditions, we also conducted synthetic experiments with narrower prior distributions for $\theta_{\mathrm{init}}$ and $q_{\mathrm{init}}$ ($0.06$ K and $0.03$ g kg$^{-1}$), labeled 'narrow init' below.

Note that we effectively use a so-called weakly informative prior (Banner et al., 2020) to limit the need for strong background information about the parameters. Moreover, we have adopted priors that are (or can be readily be transformed to) Gaussian

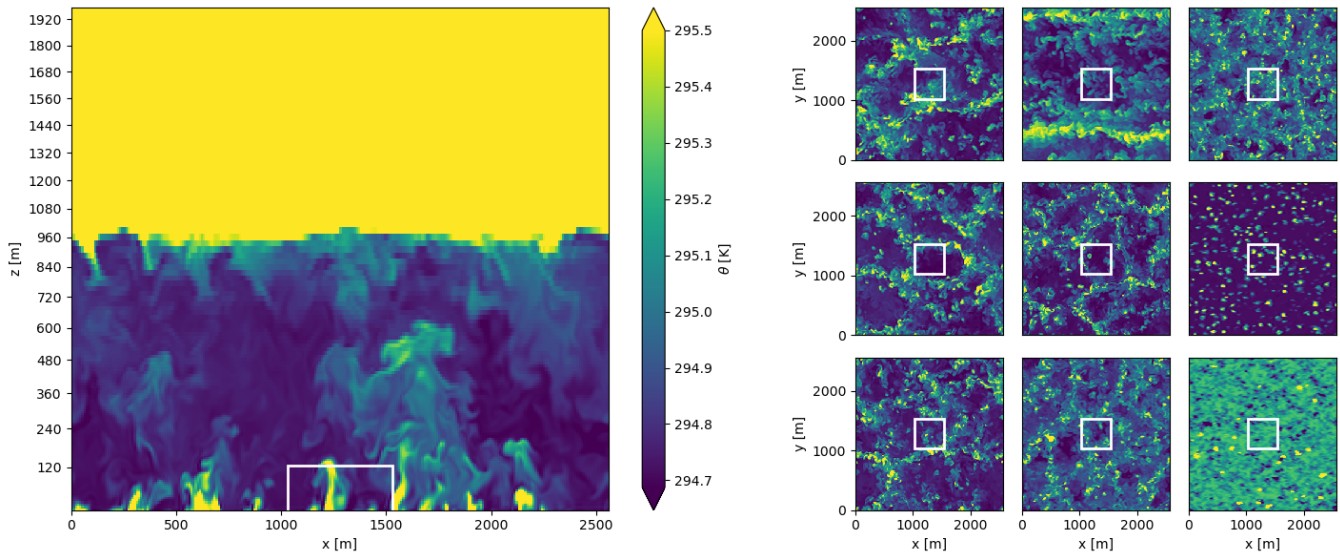

**Figure 1.** Left: Instantaneous potential temperature $\theta$ in the $x - z$ cross section at the center of the domain for the truth run at 1 h simulation time. The vertical axis is scaled with a power law function for better visibility of the boundary layer. Right: Instantaneous potential temperature $x - y$ cross section at 100 m height for the truth run (lower left panel) and eight typical ensemble members at 1 h simulation time. The color scale is independent for each ensemble member and omitted for clarity since we would like to emphasize relative differences in the spatial structure of the turbulent fields. In all plots, the white box indicates the domain for possible drone measurements.

distributions both for simplicity and because of the assumptions in the ensemble Kalman-based methods (see e.g., Carrassi et al., 2018) that we use. In theory, these methods assume a Gaussian prior and are thus closer to optimal when this assumption is satisfied. In practice, we can satisfy this assumption by using Gaussian anamorphosis techniques to transform bounded physical variables to an unbounded Gaussian space (Bertino et al., 2003; Aalstad et al., 2018). We have also included the possibility to add further information to these priors through correlations between individual parameters, because we know that in reality some conditions can make others more probable. For example, based on experience from EC flux measurements we see that a large sensible heat flux makes a large latent heat flux more likely (as necessitated by the energy balance) with typical correlations of $0.5$ to $0.8$ in data from our sites. To accommodate such prior knowledge, our framework allows us to draw correlated samples for the parameters from a joint prior distribution with a specified covariance matrix. For the analysis of the real-world drone measurements (later described), we prescribe a prior correlation between $H$ and $LE$ of $0.5$. All other prior correlations are left at $0$. For ease of interpretation in the synthetic experiments, we kept the prior correlations between $H$ and $LE$ at $0$.

### 2.1.3 Drone measurements, observations and errors

Throughout this study, small multi-rotor drones equipped with light-weight sensors for air temperature and relative humidity are used along with a flight controller that estimates the drone's tilt angle for an indirect measurement of the horizontal wind

speed $U$. This drone system is used for the real-world measurements and emulated for the synthetic experiments. A thin type-
K thermocouple connected to a high accuracy thermocouple amplifier (MCP9600, Microchip Technology, USA) is employed
to measure air temperature. A BME280 (Bosch, Germany) capacitive relative humidity sensor is used, which also measures
barometric pressure (and thus elevation). These sensors sample every 10 s, which is slower than the actual response times of
these sensors as well as the time steps of our LES runs. In practice, these sensors have slightly different response functions
with time constants of a few seconds, but for simplicity we consider the samples to represent near-instantaneous values (at
least for computing the mean observations). The data is converted to potential temperature $\theta$ and specific humidity $q$ (for which
the measurements of air pressure are used). Instead of mounting an anemometer on the drone to measure wind, we follow a
common force-balance method using the drone's tilt angle during hovering to infer the wind speed (Neumann and Bartholmai,
2015; Palomaki et al., 2017). The tilt angle is estimated by the drone's state estimator (an Extended Kalman Filter implemented
in the PX4 flight stack) based on the flight controller's (Pixhawk 4, Holybro, China) IMU sensors. Using the quadratic drag
law, the drag on the drone's body can be estimated as

$$D = mg\tan(\alpha) = \frac{1}{2}C_D\rho A v^2, \tag{4}$$

where $m$ is the mass of the drone (1.9 kg), $g$ the gravitational acceleration (9.81 m s$^{-2}$), $C_D$ the drone's drag coefficient
(estimated as 2.8 using wind tunnel experiments (Neumann and Bartholmai, 2015)), $\alpha$ the tilt angle and $\rho$ the air density (both
estimated by the flight controller), $A$ the drone's exposed area (estimated as 150 cm$^2$ from all directions), and $v$ the relative
horizontal wind speed. When the drone hovers at a fixed position, the horizontal wind speed $U$ can be assumed to be equal to
$v$. This method does not explicitly account for drag forces from rotor movements, which introduces additional uncertainty in
the wind speed estimation. We used an X500 kit (Holybro) as drone platform, which typically provides a total flight time of
15 min with the battery and payload that we employed (see photos in Figure 2).

The 10 s measurements of $\theta$, $q$ and $U$ are aggregated to mean values of all measurements taken when the drone hovers at a
fixed position (denoted as $\bar{\theta}$, $\bar{q}$ and $\bar{U}$). We additionally compute the differences between subsequent mean values to add local
mean gradients to our observations (denoted as $\Delta\bar{\theta}$, $\Delta\bar{q}$ and $\Delta\bar{U}$). This is done in a cyclic manner through the measurement
locations, so that the local gradient at the first position is calculated as the difference to the last location.

The assumed error statistics of these observations are based on noise in the measurements caused by sensor imperfections and
the mismatch between the scale of the observation and the scale of the model (representativeness error, see van Leeuwen, 2015),
which is typical in meteorological data (Gandin, 1988). The related spatio-temporal representativeness errors are affected by
the rotor wash from the drone that mixes the air around the drone and makes its measurements more representative for spatial
scales similar to the LES grid spacing. We first estimate the measurement error of the 10 s samples and then calculate the
corresponding observation error by scaling the standard deviation of the (near) instantaneous measurement error with the
inverse square root of the number of samples that are temporally aggregated to an observation. Based on the central limit
theorem (e.g. Chopin and Papaspiliopoulos, 2020), the error model assumes this observation error to be Gaussian, independent,
and uncorrelated between different variables, which corresponds to a diagonal observation error covariance matrix. Systematic
errors that occur for error distributions that are asymmetrically distributed with respect to zero, are assumed to be negligible.

This leads to the following definition for the diagonal observation error covariance matrix $\mathbf{R} \in \mathbb{R}^{d \times d}$ employed in this study

$$\mathbf{R} = \text{diag}\left(\boldsymbol{\tau} \odot \boldsymbol{\sigma}^2\right),\tag{5}$$

where $\text{diag}(\cdot)$ is the diagonal operator that converts a vector to a diagonal matrix, $\boldsymbol{\tau} \in \mathbb{R}^d$ is a scaling vector, $\odot$ denotes the element-wise product, $\boldsymbol{\sigma} \in \mathbb{R}^d$ contains the measurement error standard deviation for each observation. The elements of the scaling vector are defined as follows

$$\tau_i = \begin{cases} 1/S & \text{if mean,} \\ 2/S & \text{if local mean gradient,} \end{cases}\tag{6}$$

where $S$ is the number of measurement samples that are averaged to form an observation. As elaborated in Section 2.2, we test two types of flight plans. The first type involves step-wise vertical profiles while the drones hover in place for a 2 minute averaging period with a 10 s sampling interval such that $S = 12$. The second type involves random exploration where no averaging is performed such that $S = 1$. In summary, following independent Gaussian error propagation, this observation error covariance matrix implies that observation errors are uncorrelated, decrease with number of samples $S$ in an averaging period, and are larger for local mean gradients than for means.

The elements of $\boldsymbol{\sigma}$ are determined by the measurement error standard deviation of the respective sensors. For temperature measurements on drones, observational errors stem from radiative and adiabatic heating (due to air pressure fluctuations around the drone), and typical absolute root-mean-square errors are in the range $0.2$ to $0.3$ K (Wildmann et al., 2013). Here, we assume a standard deviation of $0.3$ K for the measurement error. The standard deviation of the measurement error for $q$ is estimated as 3% relative humidity (corresponding to about $0.1$ g kg$^{-1}$ specific humidity at typical air temperatures) that is based on the stated accuracy of the capacitive humidity sensor (BME280). For the horizontal wind speed $U$, the standard deviation for the measurement error is conservatively estimated to be $2.0$ m s$^{-1}$. Other studies using Inertial Measurement Unit data of multi-copter drones for wind estimation report measurement uncertainties of less than $0.5$ m s$^{-1}$ (Palomaki et al., 2017), but since we did not evaluate this uncertainty for our drones, we decided to use a somewhat larger value to avoid underestimating this uncertainty.

### 2.1.4 Data assimilation schemes

We implemented four data assimilation schemes and assess their performance for the problem at hand (i.e. inference of $H$ and $LE$). We used $N_e = 100$ model realizations (referred to as 'ensemble members' or 'particles' in data assimilation) each with a different set of parameter values, to represent the prior probability distribution.

For the first scheme, the particle batch smoother (PBS) introduced by Margulis et al. (2015) is used. The PBS is a batch-smoother version of the particle filter that is widely used in the snow data assimilation community (Fiddes et al., 2019; Alonso-González et al., 2021). It is effectively a particle filter without resampling, tantamount to basic sequential importance sampling. This scheme is obtained by using a particle representation, i.e. mixture of Dirac delta functions $\delta(\cdot)$, of the prior which serves as the proposal distribution to perform importance sampling-based Bayesian inference as outlined in Appendix A. The resulting

posterior is effectively a weighted sum of particles $p(\mathbf{x}|\mathbf{y}) = \sum_{i=1}^{N_e} w_i \delta(\mathbf{x} - \mathbf{x}_i)$ where the weights are given by the likelihood ratio

$$w_i = \frac{\exp\left(-\frac{1}{2}\boldsymbol{\varepsilon}_i^{\mathrm{T}}\mathbf{R}^{-1}\boldsymbol{\varepsilon}_i\right)}{\sum_{k=1}^{N_e}\exp\left(-\frac{1}{2}\boldsymbol{\varepsilon}_k^{\mathrm{T}}\mathbf{R}^{-1}\boldsymbol{\varepsilon}_k\right)}, \tag{7}$$

in which $(\cdot)^{\mathrm{T}}$ denotes the transpose and $\boldsymbol{\varepsilon}_i = \mathbf{y} - \widehat{\mathbf{y}}_i$ are the predicted observation errors where $\widehat{\mathbf{y}}_i = \mathcal{G}(\mathbf{x}_i)$ are the predicted observations from the forward model based on parameters that have been drawn from the prior $\mathbf{x}^{(i)} \sim p(\mathbf{x})$. It has been shown that the direct application of basic particle methods (i.e., importance sampling using the prior as the proposal) such as this often does not work well in high-dimensional systems (Snyder et al., 2008), but several more sophisticated variants are shown to have potential to overcome this limitation (van Leeuwen et al., 2019).

For the second scheme, the classic (stochastic) version of the ensemble smoother (ES) that involves perturbing the observations (van Leeuwen and Evensen, 1996; Burgers et al., 1998) is implemented. Although van Leeuwen (2020) recently showed that to be consistent with Bayesian theory, this stochastic scheme should perturb the predicted (i.e. modeled) observations rather than the actual observations, this does not have any practical impact on the results due to the symmetric nature of the Gaussian perturbations. The ES scheme is a fixed-interval batch smoother version of the widely used ensemble Kalman filter (EnKF; Evensen, 1994) that assimilates all observations simultaneously in a batch rather than sequentially. Such batch assimilation is more suitable for the inverse problem pursued herein (Evensen, 2018). Ensemble Kalman Filtering (EnKF) methods are successfully used in data assimilation applications in meteorology and oceanography with tens of millions of dimensions (Carrassi et al., 2018). While the EnKF assumes that the forward model is linear and that all distributions are Gaussian, it turns out that the EnKF is robust to deviations from these assumptions in many applications (Katzfuss et al., 2016). These methods and the underlying equations are described in Appendix B.

For the third scheme, we use the Ensemble Smoother with Multiple Data Assimilation (ES-MDA) (Emerick and Reynolds, 2013). The ES-MDA is an iterative ensemble smoother that has been suggested as a more viable alternative to the non-iterative ES for highly non-linear forward models. In this iterative scheme, the same data is assimilated multiple times with an inflated observation error covariance matrix to better handle the non-linear mapping between the parameters of interest and the observations. In particular, the gradual updating reduces the impact of the linear assumption in the ES update step. Despite using the data more than once, this iterative scheme remains consistent in a Bayesian sense since inflation is performed in such a way that the iterations are equivalent to assimilating the data once with a linear model. At the root of these iterative schemes we find the idea of tempered transitions, which is a technique that is widely used in challenging Bayesian inference tasks (Neal, 1996; Stordal and Elsheikh, 2015; Iglesias and Yang, 2021). This tempering, in combination with their derivative-free implementation, has placed iterative ensemble Kalman methods at the frontier of ongoing research in Bayesian inverse problems (Stuart, 2010; Iglesias et al., 2013; Schillings and Stuart, 2017) which is helping to both formalize, improve, and generalize this family of methods (Garbuno-Inigo et al., 2020; Iglesias and Yang, 2021; Cleary et al., 2021; Dunbar et al., 2022a). The equations and workflow for the ES-MDA scheme used herein are presented in Appendix B.

As a fourth scheme, a combination of the schemes described above is developed and implemented in a Particle-adjusted Iterative Ensemble Smoother (PIES). The PIES scheme is obtained by using the output of an iterative ensemble smoother, i.e.

a Gaussian distribution, as the proposal distribution in importance sampling as outlined in Appendix A. Herein, we use the estimated Gaussian distribution from the penultimate iteration of the ES-MDA scheme as the proposal distribution. This new PIES scheme is an adaptation of the weighted EnKF described elsewhere (Papadakis et al., 2010) and the iterative ensemble smoothers. As with the PBS, the resulting posterior is effectively a weighted sum of particles $p(\mathbf{x}|\mathbf{y}) \simeq \sum_{i=1}^{N_e} w_i \delta(\mathbf{x} - \mathbf{x}_i)$ with weights given by

$$
w_i = \frac{\exp\left(-\frac{1}{2}\boldsymbol{\varepsilon}_i^{\mathrm{T}}\mathbf{R}^{-1}\boldsymbol{\varepsilon}_i - \frac{1}{2}\widetilde{\mathbf{x}}_i^{\mathrm{T}}\boldsymbol{\mathcal{C}}^{-1}\widetilde{\mathbf{x}}_i + \frac{1}{2}\widehat{\mathbf{x}}_i^{\mathrm{T}}\widehat{\boldsymbol{\mathcal{C}}}^{-1}\widehat{\mathbf{x}}_i\right)}{\sum_{k=1}^{N_e}\exp\left(-\frac{1}{2}\boldsymbol{\varepsilon}_k^{\mathrm{T}}\mathbf{R}^{-1}\boldsymbol{\varepsilon}_k - \frac{1}{2}\widetilde{\mathbf{x}}_k^{\mathrm{T}}\boldsymbol{\mathcal{C}}^{-1}\widetilde{\mathbf{x}}_k + \frac{1}{2}\widehat{\mathbf{x}}_k^{\mathrm{T}}\widehat{\boldsymbol{\mathcal{C}}}^{-1}\widehat{\mathbf{x}}_k\right)}, \tag{8}
$$

where $\widetilde{\mathbf{x}}_i = \mathbf{x}_i - \boldsymbol{\mu}$ are the anomalies from the prior mean ($\boldsymbol{\mu}$), $\mathcal{C}$ is the prior covariance matrix, $\widehat{\mathbf{x}}_i = \mathbf{x}_i - \widehat{\boldsymbol{\mu}}$ are the anomalies from the mean of the penultimate ES-MDA iteration ($\widehat{\boldsymbol{\mu}}$), $\widehat{\boldsymbol{\mathcal{C}}}$ is the covariance matrix from the penultimate ES-MDA iteration, and the particles have been sampled from the Gaussian distribution estimated from the penultimate ES-MDA iteration such that $\mathbf{x}_i \sim \mathrm{N}(\widehat{\boldsymbol{\mu}}, \widehat{\boldsymbol{\mathcal{C}}})$ with predicted observations $\boldsymbol{\varepsilon}_i = \mathbf{y} - \widehat{\mathbf{y}}_i$ with $\widehat{\mathbf{y}}_i = \mathcal{G}(\mathbf{x}_i)$. Importance sampling is more effective the closer the proposal is to the target posterior distribution (MacKay, 2003). So in theory it would be better to use the posterior estimate from the final (rather than penultimate) iteration of the ES-MDA for the proposal in PIES, but this would come at a high computational cost of requiring an additional round of runs of the LES ensemble. The motivation for pursuing the PIES scheme is that the ES-MDA produces a biased approximation of the posterior for non-linear forward models (Stordal and Elsheikh, 2015). Although this bias is typically less severe than that of non-iterative ensemble Kalman methods (Emerick and Reynolds, 2013), it would nonetheless be advantageous to find efficient methods to reduce it. PIES is a straightforward translation of the scheme of Papadakis et al. (2010) to iterative ensemble smoothers such as the ES-MDA. As such, PIES can be viewed as a simple extension of the ES-MDA that does not necessarily impose any noticeable computational burden and might improve performance. As with all particle methods, the effective sample size can be used to diagnose degeneracy in the ensemble of particles (Chopin and Papaspiliopoulos, 2020). A low ($\ll N_e$) effective sample size indicates degeneracy due to the fact that the proposal is too far from the target posterior.

## 2.2 Synthetic experiments

To compare the performance of different DA schemes and observation strategies, a set of so-called synthetic (or twin) experiments were conducted. The experiments were performed by extracting synthetic measurements from one forward model run, labeled 'truth', where the true values of each parameter are assumed to be known. These measurements were then intentionally corrupted with noise based on the assumed measurement error model, and converted to drone-based observations of mean values and local mean gradients. The true values of the six parameters were chosen to represent typical summertime conditions during daytime above high-latitude wetlands with a sparse tree cover ($H = 160$ W m$^{-2}$, $LE = 120$ W m$^{-2}$, $z_0 = 0.25$ m, $\theta_{\mathrm{init}} = 294.1$ K, $q_{\mathrm{init}} = 5.55$ g kg$^{-1}$, $u_g = 1.5$ m s$^{-1}$). Experiments with higher wind speed using $u_g = 6.0$ m s$^{-1}$ were also performed to test how increased mixing from mechanical turbulence (as opposed to buoyancy-driven turbulence) and the correspondingly reduced spatial gradients affect the drone data assimilation flux estimates.

To evaluate the performance of the DA schemes, we use standard point metrics such as the root-mean-square error (RMSE) and bias (mean error) of the ensemble medians with respect to the true values. To also measure the quality of the entire ensemble, we employ the continuous ranked probability score (CRPS; Hersbach, 2000), which is a widely used score for ensemble verification in numerical weather prediction that generalizes the mean absolute error to an ensemble. It measures the distance between the entire ensemble and a deterministic reference value, in our case the truth, with 0 being the best possible score that only occurs for a degenerate ensemble centered on the truth. To quantify the overall information gain in an experiment, we also calculate a Kullback-Leibler divergence (KLD; see e.g. Murphy, 2022) that measures the distance between the posterior and prior distribution (Perez-Cruz, 2008). We use nats as a unit for information content, where 1 nat corresponds to the information content of an event when the probability of that event occurring is $1/e$. These four metrics quantify different aspects of the fit and information gain of parameter distributions and can hence give a more holistic evaluation of a synthetic experiment.

Two different types of flight plans were used to generate the observations, both adhering to most countries' legal constraints that drones must not fly above altitudes of 120 m and that they must stay within visual range (a lateral domain estimated to be 500 m by 500 m). Based on MOST, we expect mean vertical gradients in measurements to increase towards the surface. For the first type of flight plan, we thus used a step-wise vertical profile with step sizes that increased with altitude. In particular, given the limited flight time of small multi-rotor drones, we used six vertical levels (at 10, 20, 30, 50, 70, and 100 m) with the drone hovering in place at each level for 2 minutes. We also performed synthetic experiments with flight times of 24 minutes, flying this step profile twice. The second type of flight plan tested explores a larger spatial domain instead of hovering at a fixed position for 2 min. We implemented this approach using a random walk with biased directionality that is based on movement models used for biological systems (Codling et al., 2008). Here, every 10 s the drone can stay at its position or move 20 m laterally ($x$ or $y$ dimension) and/or 10 m vertically ($z$ dimension), i.e. moving two LES grid cells. These moves are random, but to explore a larger space, the probability to continue moving (or staying) in the same direction for each spatial dimension is 0.8, compared to a probability of 0.1 for the other two options. Examples of these random exploration flights are shown in Figure S2 (see Supplementary material). For this random exploration, the instantaneous (10 s) measurements are assimilated as observations.

We also include the possibility of using multi-drone observations to test the performance of a mobile sensor network on a drone swarm. For this purpose, we assume individual drones to be identical in sensor specifications and flight time corresponding to a so-called homogeneous swarm (see e.g. Ferreira-Filho and Pimenta, 2019).

## 2.3 Field experiments

Field campaigns at two ecohydrological research sites with different climatic conditions were conducted: a boreal wetland in south eastern Norway (Hisåsen, 61.11°N, 12.26°E, elevation 680 m a.s.l., mean annual air temperature 2.7°C at the closest weather station) and a palsa mire in the discontinuous permafrost zone in northern Norway (Iškoras, 69.34°N, 25.30°E, 355 m a.s.l., mean annual air temperature −1.6°C at the closest weather station). Figure 2 shows photos of the two sites to give an impression of the settings.

These sites feature independent flux measurements from EC systems installed at 2.8 m a.g.l. at both sites. A CSAT3 sonic anemometer (Campbell Scientific) was used at Iškoras and a HS50 (Gill) at Hisåsen. Both sites use an Li-7200 infra-red gas

analyzer (Li-Cor) for $H_2O$ mixing ratios. Raw data from these instruments are sampled at 20 Hz and processed to 30-min flux data following the FluxNet convention implemented in EddyPro version 6.2.0 (Li-Cor). We use block average Reynolds decomposition to extract turbulent fluctuations, an anemometer tilt correction by double rotation, a constant time lag compensation, and a high and low-pass filter correction (Moncrieff et al., 2005, 1997). For quality control, the flagging system proposed in FluxNet (Foken and Wichura, 1996) was adopted and only flux data with the highest quality (flag 0) was used

here. A drone flight is only considered successful if the EC fluxes of the 30-min interval that contains the drone takeoff time meet this quality condition. Along with the EC fluxes, EddyPro also estimates their random error (the variance of the flux covariance) due to sampling errors that arise from the small number of large eddies that dominate the flux during typical sampling periods following Finkelstein and Sims (2001).

One field campaign was conducted at the Iškoras site in July 2020, resulting in two successful drone flights. At Hisåsen,

intensive campaigns were carried out in June 2020 with 12 successful flights, in October 2020 with one successful flight, and in September 2021 with three successful flights. An overview of the conditions and EC fluxes at Hisåsen, June 2020 (see Figure S3 in Supplementary material) shows that EC fluxes have best quality flags during daytime with random flux error estimates of around 10 W m$^{-2}$.

We used the vertical step profile flight plan in all these flights. As these drone measurements are taken at altitudes up to

100 m a.g.l., the resulting flux estimates represent a larger surface footprint area (kilometer scale) compared to the EC method (tens to hundreds of meters). At both field sites, the footprint of the EC tower is dominated by wetlands, while the larger-scale surroundings feature forested areas with potentially different turbulent heat flux characteristics.

## 3 Results

### 3.1 Synthetic experiments

Figure 3 shows the prior and estimated posterior distributions for a synthetic experiment with observations from one drone flying a step profile for 12 min. The PBS and PIES schemes tend to assign most weight to only a few ensemble members. These almost degenerate posterior distributions are therefore visualized by their central 95$^{th}$-percentile range instead of their kernel density estimated probability density function in Figure 3. Both the ES and ES-MDA yield a posterior with a constrained spread that is approximately centered at the truth. In this experiment, there is a marked information gain from prior and posterior

of both the ES and the ES-MDA scheme, with KLDs of 2.9 and 3.6 nat, respectively. The KLDs for PBS and PIES are 4.6 and 6.5 nat, respectively, but the posterior distributions are practically degenerate with effective sample sizes of 1.0 and 1.2, respectively. The marginal distributions of both sensible heat flux $H$ and latent heat flux $LE$ show a considerable update towards their true values when moving from the prior to the posterior distribution, especially for the ES-MDA as evidenced by a low continuous ranked probability score of about 12 W m$^{-2}$ in this synthetic experiment. In line with the KLD results, the

ES-MDA gives a smaller posterior spread compared to the ES, with a standard deviation of 37 W m$^{-2}$ for $H$ and 46 W m$^{-2}$

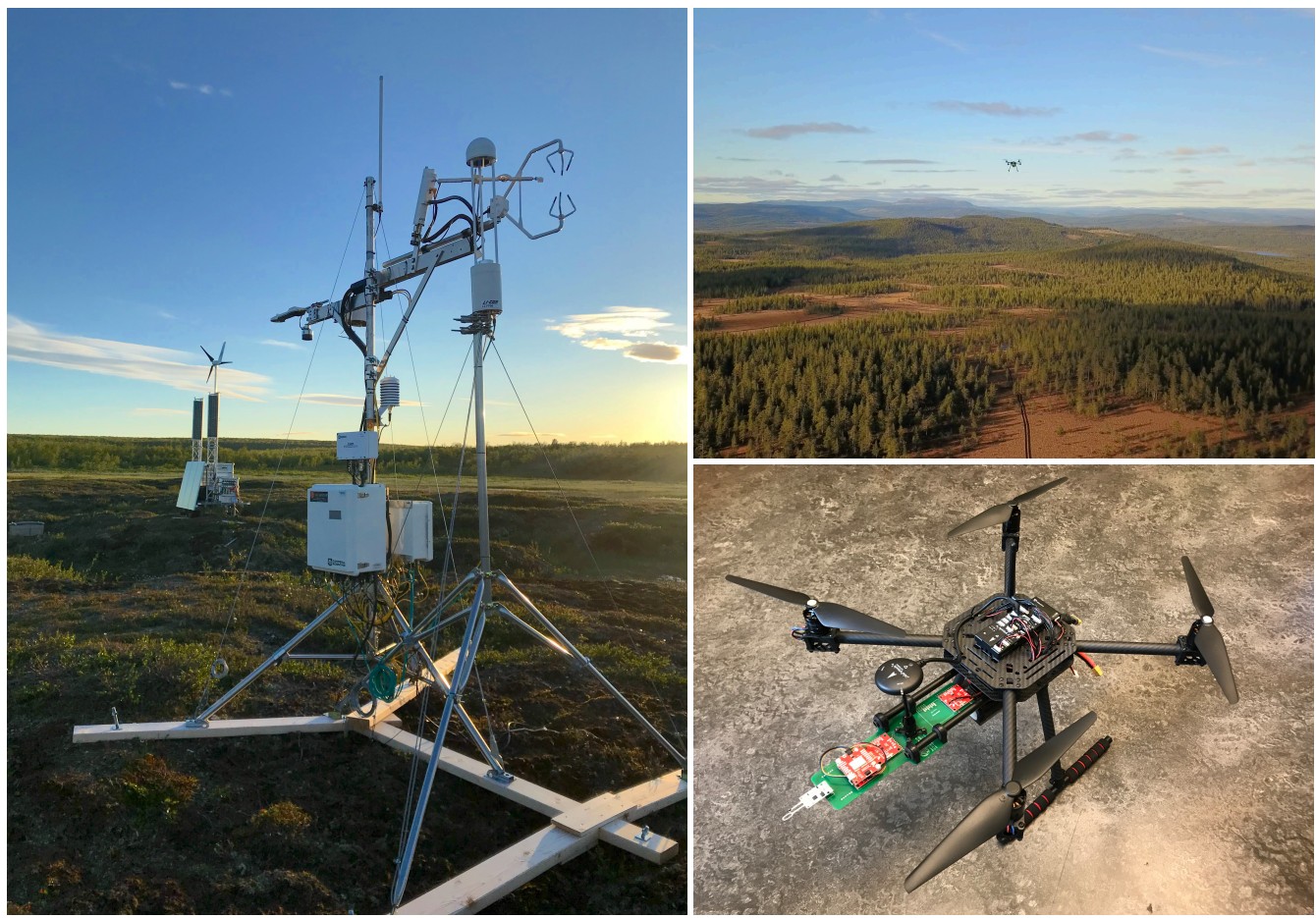

**Figure 2.** Examples of field sites and equipment. Left: Eddy covariance tower at the Iškoras palsa mire. Top right: Drone above the Hisåsen site (photo by Pierre-Marie Lefeuvre). Bottom right: Drone with sensors.

for $LE$. The slightly wider posterior spread for $LE$ is expected due to the relatively large observational error for specific humidity, which contains most information about $LE$. We see that the drone observations do not include much information to constrain the roughness length $z_0$, as this nuisance parameter appears to be hardly updated. This lack of adjustment of $z_0$ can be explained by the relatively large observational error associated with the wind speed estimates. Hence, our prior belief strongly governs the distribution of this parameter. It is nonetheless important to account for uncertainty in this nuisance parameter so as to avoid over-confident and possibly wrong inferences about the fluxes of interest. External information, for example from remotely sensed land cover data, may help constrain this parameter (see Section 4.2). The two parameters for the initial conditions update slightly towards the true values of this synthetic experiment. We see a noteworthy equifinality issue (Beven, 2006) related to the initial conditions in the given problem: In the posterior parameter estimates, there is a negative correlation between the parameters $H$ and $\theta_{\mathrm{init}}$ ($R = -0.89$), as well as between $LE$ and $q_{\mathrm{init}}$ ($R = -0.80$). This

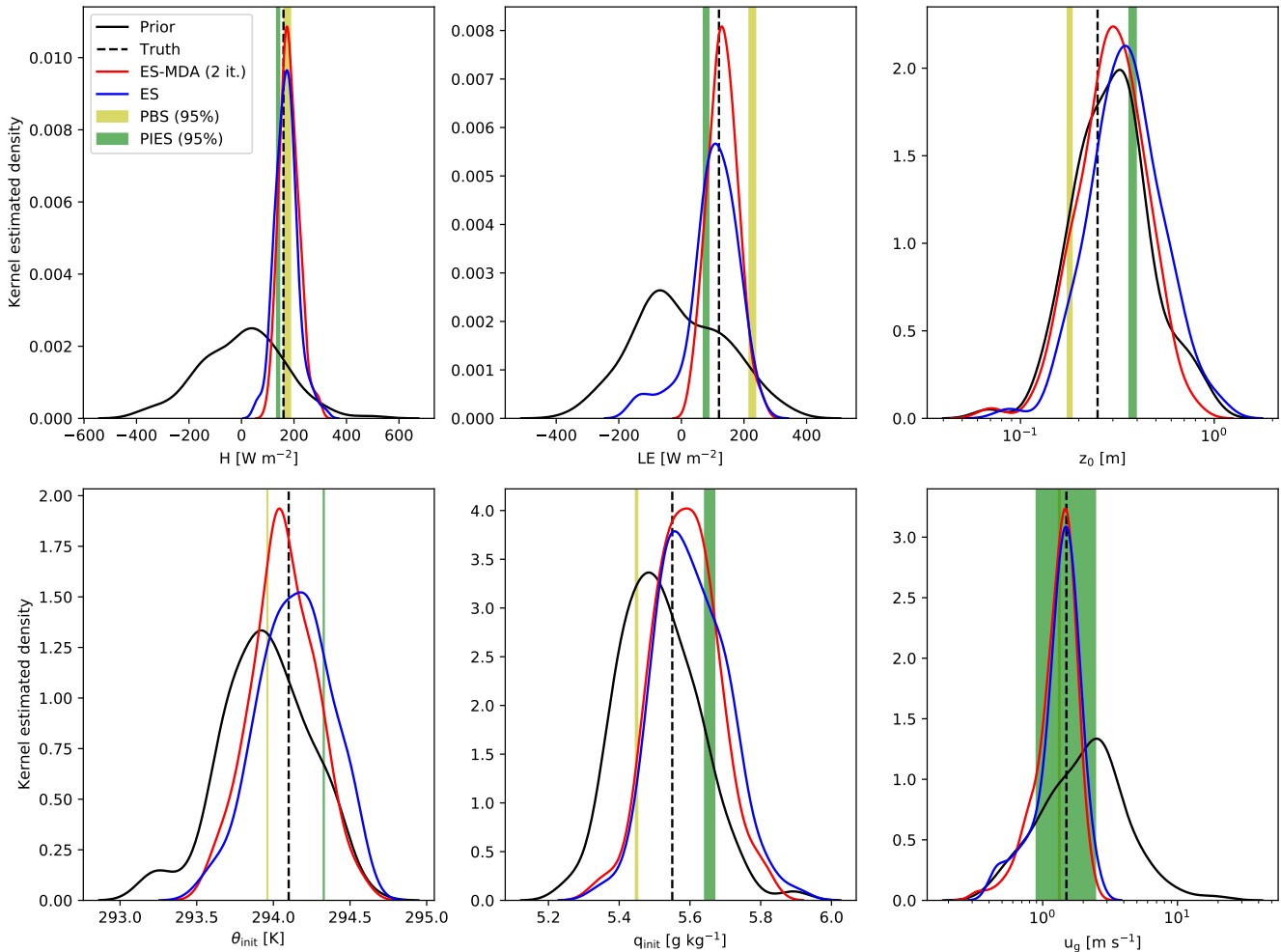

**Figure 3.** Marginal parameter distributions for the prior (black) as well as the ES-MDA (with $N_a = 2$ iterations, red), ES (blue), PBS (yellow shading shows the central $95^{th}$-percentile range), and PIES (green shading shows the central $95^{th}$-percentile range) posterior estimates along with the location of the truth (black dashed vertical line) in a synthetic experiment with one drone flying a step profile for 12 min.

negative correlation suggests that an ensemble member with low initial temperature and large sensible heat flux gives similar temperature predictions as an ensemble member with high initial temperature and small sensible heat flux. While this may not be a surprising result, at least a-posteriori (or with hindsight), it emphasizes the importance of initial conditions for drone-based surface flux estimations. Despite the relatively large observational error associated with the wind speed estimates, we see that

$u_g$ updates considerably towards the truth for all DA schemes.

Varying the sampling strategies (step profile vs random exploration), flight time (12 vs 24 minutes), number of drones (1 vs 5), uncertainty in initial conditions (narrow vs broad), and the geostrophic wind speed (1.5 vs 6.0 m s$^{-1}$) led to a total of 16 synthetic experiments. Table 1 compares the four DA schemes with respect to their evaluation metrics across all synthetic

**Table 1.** Average evaluation statistics across all 16 synthetic experiments for different DA schemes.

| Scheme | RMSE [W m$^{-2}$] | | Bias [W m$^{-2}$] | | CRPS [W m$^{-2}$] | | KLD(post.\|\|prior) |
|---|---|---|---|---|---|---|---|
| | $H$ | $LE$ | $H$ | $LE$ | $H$ | $LE$ | [nat] |
| PBS | 70.0 | 90.8 | -49.8 | -38.9 | 70.2 | 81.0 | 4.6 |
| ES | 36.4 | 73.7 | 13.7 | -41.9 | 21.6 | 47.2 | 5.9 |
| ES-MDA | 29.2 | 43.0 | 4.3 | -18.3 | 18.6 | 28.6 | 7.7 |
| PIES | 44.9 | 64.0 | -7.7 | -40.1 | 31.2 | 52.8 | 7.4 |
| Prior | 144.2 | 153.1 | -144.2 | -153.1 | 92.3 | 77.3 | 0.0 |

experiments. The error-based evaluation metrics, i.e. the RMSE, bias, and CRPS, indicate that the ES-MDA scheme performs
best. For example, the ES-MDA provides a mean fractional improvement in RMSE of 76% relative to the prior, which is
considerably higher than the other schemes with values at 64% (PIES), 64% (ES), and 46% (PBS). The ES-MDA scheme
gives the largest information gain from the prior to posterior, as indicated by its KLD.

Figure 4 shows the comparison of the ES-MDA (with two iterations) posterior distributions for $H$ and $LE$ for our set of
synthetic experiments. Attention is drawn to the posterior spread and whether the true flux values are encompassed by the
posterior distributions. For the case of one drone flying one step profile, we see comparable results for high and low wind
speeds. Recall that the experiment with $u_g = 1.5$ m s$^{-1}$ corresponds to the experiment shown in Figure 3. As expected,
increasing the flight time from 12 to 24 min, i.e. flying two consecutive step profiles, narrows the posterior distributions (but
by less than a factor of two). Using observations from five drones flying step profiles, the posterior distributions become even
narrower while still encompassing the 'truth'. A similar reduction in flux uncertainty is achieved by the narrower priors for the
initial conditions $\theta_{\mathrm{init}}$ and $q_{\mathrm{init}}$. Using the biologically inspired random exploration flight strategy, we find that several of the
posterior distributions can become narrower than their step-profile counterparts, but do not always include the true flux value,
which is a symptom of the ES-MDA assimilation results being overconfident. This effect can be related to random exploration
containing more observations (but without aggregation of multiple measurements) compared to step profiles.

## 3.2 Field experiments

Figure 5 shows an example of the observed field data and the posterior predicted LES data by the ES-MDA scheme from
one flight at the Iškoras site. Both the drone observations and ensemble posterior predictions show an increase in potential
temperature and specific humidity towards the surface, indicating a positive $H$ and $LE$. Due to the friction at the surface, wind
speeds tend to decrease at lower altitudes and roughly follow the characteristic logarithmic mean wind profile (modified by the
stability effects of the given temperature stratification) that MOST predicts for average values. The measured mean values and
local mean gradients are generally well reproduced by the posterior LES ensemble. There is a small tendency that the measured
profiles have stronger vertical gradients in the lowest 30 m than the LES ensemble, possibly indicating an effect at the field site
that is not included in LES runs such as surface heterogeneity in sources and sinks as well as roughness elements.

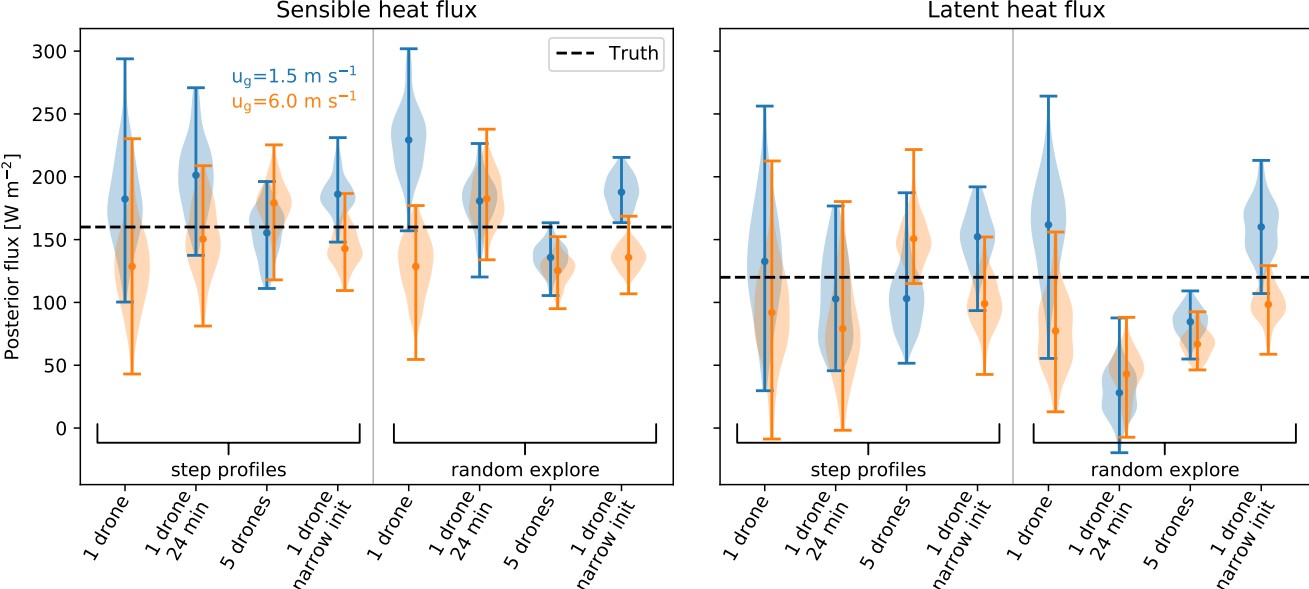

**Figure 4.** Assessment of observation strategies from synthetic experiments with true $H = 160 \text{ W m}^{-2}$ and $LE = 120 \text{ W m}^{-2}$. Violins show the Kernel density estimated posterior distributions of surface sensible (left) and latent (right) heat fluxes obtained by the ES-MDA method with two iterations. The caps of the violins mark the extrema of the ensemble and the dots the mean values. Colors denote two different wind speeds $u_g$. The experiment with one drone flying a step profile corresponds to the case shown in Figure 3. Flight plans for the different observation strategies are shown in Figure S2 in the Supplementary material.

Figure 6 shows the surface flux comparisons with EC from all the field experiments. For the sensible heat flux, good agreement between the two methods is noted at both sites with a high correlation coefficient ($R = 0.90$). The drone data assimilation flux estimates tend to yield higher fluxes than the EC method, which might be a real effect given the different footprints of the two methods (i.e. wetlands dominating the EC footprint have lower $H$ and higher $LE$ than their surroundings). For the latent heat flux estimates, the drone data assimilation typically yields larger flux uncertainty and the correlation with EC fluxes is only $0.40$. There is a particularly large deviation between the methods in the three flights with the largest $LE$ flux estimates from EC. Again, this deviation could be due to real flux differences between the wetland-dominated vicinity of the EC tower and the surrounding forest. We also emphasize that the EC fluxes do not necessarily represent the 'truth' in this comparison, even though we only used EC estimates with the highest quality flags. For the sum $H + LE$, which is constrained by the available energy at the surface and may therefore be more homogeneous, the close agreement (RMSE=$58 \text{ W m}^{-2}$) and high correlation coefficient is noteworthy ($R = 0.83$). As the drone-DA uncertainty estimates are largely a result of our experimental design (flight time, sensor noise, etc.) and the used prior distributions, all 18 flights show largely the same epistemic uncertainty. For the EC estimates, error bars in Figure 6 only indicate the absolute aleatoric uncertainty due to sampling limitations, which is expected to increase with flux magnitude (Finkelstein and Sims, 2001).

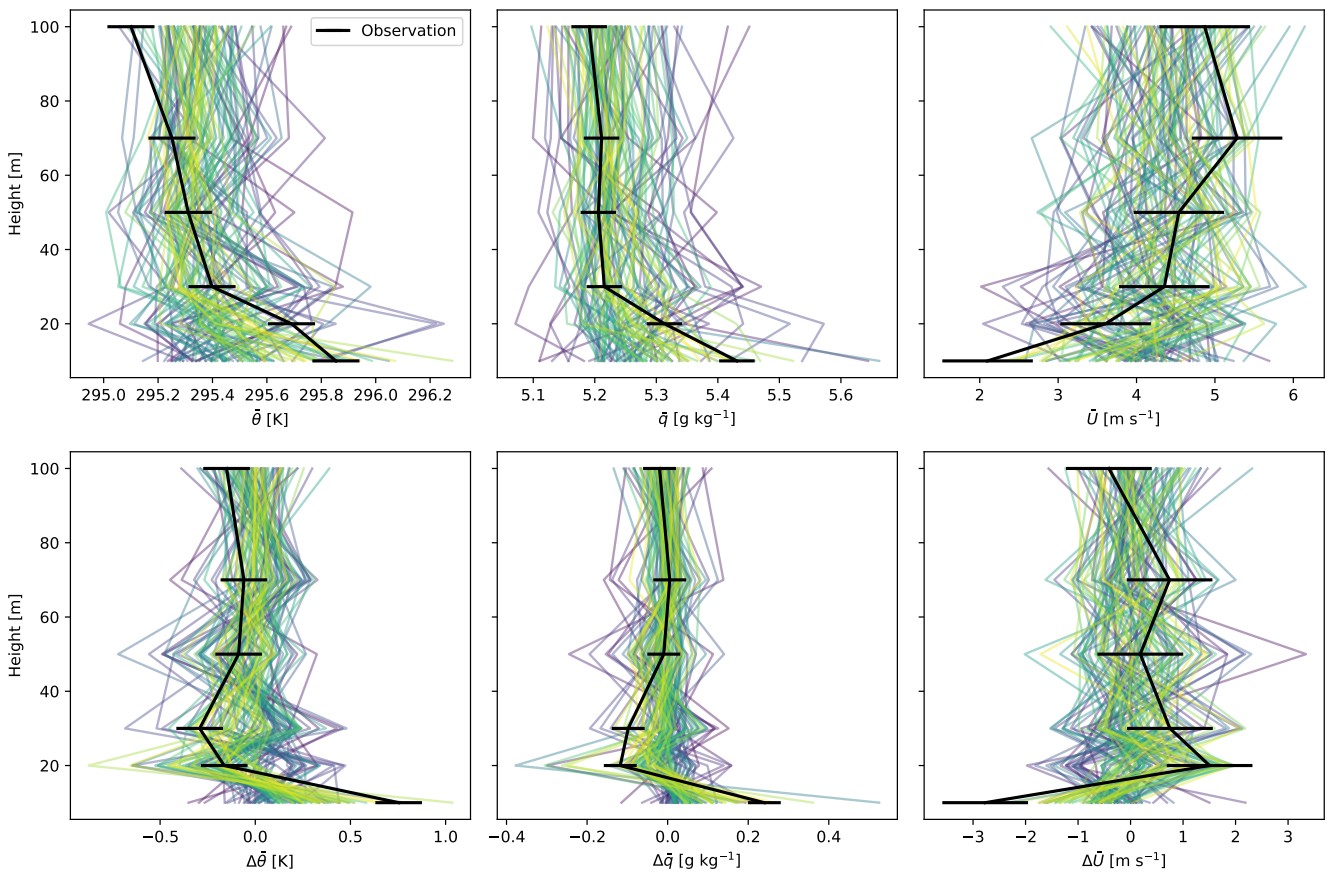

**Figure 5.** Drone observations and posterior ensemble predictions from the ES-MDA for flight 1 of the Iškoras campaign, a step profile on 2020-07-27 with takeoff at 15:20 local time. The upper panels show the successive 2-min mean values, whereas the lower panels show the local mean gradients. The line colors of the vertical profiles for the $N_e = 100$ posterior ensemble members from the ES-MDA correspond to their log-likelihood with more likely values in yellow and less likely values in blue. The prior predictions are not shown, because their range is so wide that one could not see any details in the posterior profiles.

## 4    Discussion

### 4.1    Potential and limitations of drone data assimilation

In this study we show how ensemble-based data assimilation of drone observations in an LES model can realistically estimate

homogeneous and stationary surface energy fluxes. Using the same assumptions as the EC technique, we find acceptable agreement of these two independent methods under field conditions where the assumptions are only approximately fulfilled, particularly for the sensible heat flux. The agreement of the methods is worse for the latent heat flux, especially at high EC estimates of $LE$. Since wetland $LE$ is known to increase more than forest $LE$ with increasing vapor pressure deficit (Helbig

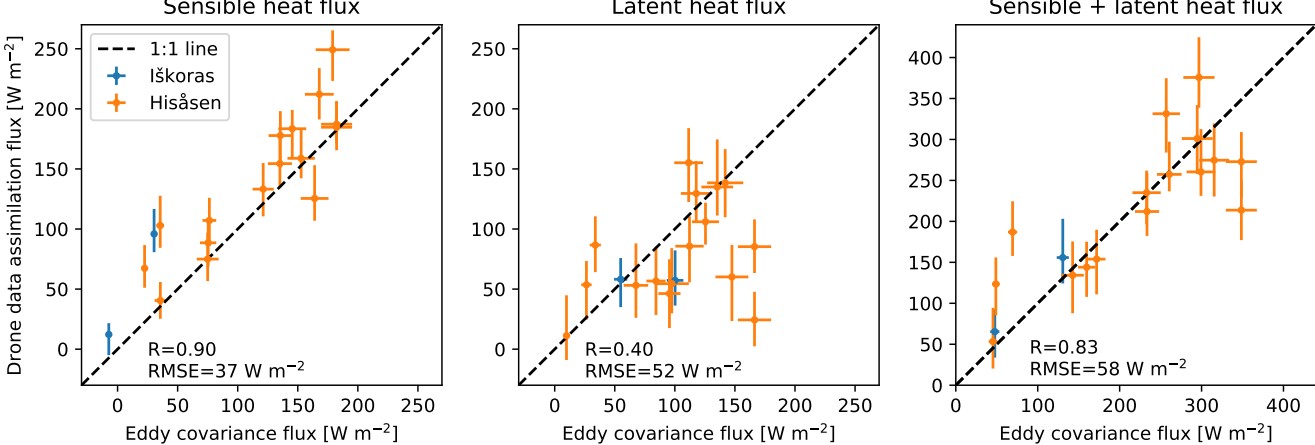

**Figure 6.** Comparison of flux estimates by EC towers and drone data assimilation as estimated by posterior distributions of the ES-MDA method with two iterations, for in total 18 separate flights at two different sites. Error bars for drone data assimilation fluxes indicate the interquartile range, and points indicate the median value. Error bars for EC fluxes indicate the random flux error estimated by EddyPro.

et al., 2020), this deviation could be due to the larger footprint of the drone data assimilation method covering more of the wetland-forest mosaic at our sites. This hypothesis could be tested in a future study with additional measurements using scintillometers, which yield fluxes that are representative for larger areas than EC. The new method operates over a large range of heat fluxes and wind speeds, but it remains to be tested how strongly stable conditions (e.g. during nighttime and/or winter) affect the technique.

The basis for these results is that drone measurements and LES capture variability at approximately the same spatial and temporal scales. During hovering, the drone position is kept fixed to within about 1 m based on GPS and barometer measurements. The drone's rotor wash, however, creates local mixing of air so that the temperature and humidity measurements represent a volume average around the drone, with a scale similar to the LES grid resolution. Incidentally, the LES time steps are at approximately the same temporal scale as sensors response times. Any remaining representativeness errors can in future studies also in theory be accounted for formally in the data assimilation framework by using an appropriate observation operator (van Leeuwen, 2015). However, a remaining limitation is that the LES output at grid levels close to the surface is affected by the subgrid scheme and the coupling to the surface, which is typically conducted using MOST.

Using typical sensor configuration and flight times of small drones, we find a relatively large posterior spread of the surface fluxes, compared to the uncertainty estimates of the EC technique that are solely accounting for sampling errors arising from the small number of large eddies captured in the 30-min flux interval. While this may be expected given the non-linear and chaotic nature of the turbulent transport process, the uncertainty estimate here is based on explicitly stated error distributions of the observations and epistemic uncertainties in the dynamical system related to initial and boundary conditions as well as parameters. It is therefore feasible to study and reduce flux uncertainties in our framework. For example, further observing

system simulation experiments can be carried out to test the impact of sensor quality, observation strategies, and large-scale meteorological forcing (see Section 4.2).

Drone flux estimation provides a relatively low-cost and mobile complement to traditional methods like EC. Most applications of drones are currently still restricted to manual flights with a human pilot in charge of the system (see, e.g., Bassi (2020) for a discussion of European airspace regulations). For fully automated and continuous drone flux measurements, a number of engineering and legal limitations need to be overcome. Nonetheless, even under the current constraints, the resulting flux temporal snapshots cover a much larger area than the typical EC footprints and can therefore be more suitable for large scale
aggregate flux measurements. These snapshots may thus assist Earth system model improvement through targeted testing of different process formulations and parameter settings, e.g. for new plant functional types (Dagon et al., 2020) or snow schemes (Aalstad et al., 2018).

   Furthermore, it is worth highlighting that trace gas fluxes can also be estimated with this drone data assimilation technique, which would be particularly relevant for $CO_2$ and methane emissions from heterogeneous permafrost environments (Pirk et al.,
2017). Gas concentration measurements from drones are already emerging as a cost-efficient tool for the petroleum industry (Asadzadeh et al., 2022) and the agriculture sector (Daube et al., 2019), where greenhouse gas emissions can contribute to climate warming. Drone data assimilation could thus be a valuable tool to help monitor such hidden – and sometimes avoidable – emission sources.

### 4.2   Possible improvements

Improved surface flux inferences can be achieved in a number of different ways, including technical improvements during data collection, such as using higher quality sensors, more drones, and better quantifying initial and boundary conditions, as well as modifying the data assimilation framework by using a larger ensemble size to improve the Monte Carlo approximation, more assimilation cycles to better account for nonlinearity, emulators to increase the smoothness of the likelihood function (Cleary et al., 2021), and a higher spatial resolution of the LES model to reduce structural model errors. The present study shows a
choice of settings that we intuitively considered reasonable or practically possible, but more work should go into systematically exploring the many orthogonal dimensions involved in the optimal experimental design of the drone data assimilation framework.

   More effort could also go into formulating the priors, especially because some parts of the covered parameter space might in reality be physically improbable (e.g. inclusion of the energy balance). Our framework allows to add further information to
the priors through correlations between individual parameters, which we only used for $H$ and $LE$ in our field experiments. The effect of these prior parameter correlations was mostly a slightly more effective exploration of the parameter space, but future studies could investigate how this feature can be used to reduce the computational costs with expensive models like LES. Other parts of the parameter space could be constrained based on independent information, for example, by using downscaled reanalysis datasets that combine other Earth observation data (e.g. Fiddes et al., 2019; Alonso-González et al.,
2021). A complementary approach could also be to directly incorporate land cover information, e.g. from satellite retrievals (Aalstad et al., 2020), into the design of flux maps in the turbulence simulation as was done in van der Valk et al. (2022). The

boundary layer height or the height of the prescribed capping inversion has been assumed to be known herein and was thus not included as a nuisance parameter in the DA scheme. Future studies should further test the validity of this assumption and explore the sensitivity of drone flux measurements to entrainment fluxes at the top of the boundary layer, which are known to affect turbulent quantities in the surface layer (van de Boer et al., 2014).

An extension of the list of uncertain parameters should ideally also be accompanied with including more observational constraints. In the current study, we assimilated observations of the mean values over short periods of times, and their local mean gradients. Higher order moments (such as variances and covariances), atmospheric structure functions (Arenas and Chorin, 2006), or other features could also be used to capture more information from the drone measurements in future studies.

The flight strategy used for the collection of observations could also be optimized. The determination of the optimal sampling strategy for mobile sensors networks (giving sparse and noisy data) can more generally be regarded as an example of the exploration-exploitation dilemma (Box and Youle, 1955; Friston et al., 2015). In practice, the choice or trade-off between fewer, high quality observations and more, low quality observations becomes an active choice of the investigator. We have only tested two simple strategies: (i) what we called an intuitive approach that involved flying a vertical step profile with averaging times of two minutes, and (ii) a more exploratory approach based on directed random walks without averaging. The results indicate that both methods can constrain the surface fluxes, but random exploration without averaging multiple measurements for an observation can give biased flux results. These biases are likely due to shortcomings of the assimilation schemes used when dealing with strongly non-linear forward models rather than the sampling strategy itself, and so could be alleviated by improving the assimilation algorithms. Future studies could moreover formalize and optimize the trade-off between exploration and exploitation more specifically using a calibrate-emulate-sample framework (Cleary et al., 2021; Dunbar et al., 2022b), to determine the optimal strategy for a given flux mapping task.

While we only applied drone data assimilation to cases with homogeneous and stationary surface fluxes, and flat terrain (as a logical first step to facilitate comparability with EC), it is clear that the method can also explicitly account for more realistic representations of these effects. Complex terrain and flux heterogeneity can be explicitly included in the LES steering data and some field sites might even require a more detailed LES setup to account for energy flux heterogeneity (Ramtvedt and Pirk, 2022). Representing the mesoscale meteorological setting more realistically could also be achieved through a newly-developed mesoscale nesting of PALM (Lin et al., 2021). Finally, even non-stationary fluxes could be included by going from a smoothing to a filtering data assimilation framework. This might in future help to complement EC measurements and maybe even improve on them by identifying assumption-violations that are causing the energy balance closure problem of the EC method (Stoy et al., 2013).

### 4.3 Data assimilation schemes for turbulent transport

Spatio-temporal data assimilation with LES is relatively complex mathematically and not commonly studied. In this case the forward model for the data assimilation becomes highly non-linear, which violates the assumptions of commonly used methods for higher-dimensional problems, such as Ensemble Kalman filters and smoothers. To avoid biased results, the degree of violation must be reduced, which can be achieved by the iterative approach implemented in the ES-MDA scheme. Our

results, particularly Figure 3 and Table 1, show that the ES-MDA scheme can markedly outperform both the ES and the particle-based methods tested herein. These findings are in close agreement with earlier snow data assimilation experiments (Aalstad et al., 2018; Alonso-González et al., 2022) that compared these schemes with similarly (i.e. medium) sized parameter spaces, albeit with considerably less complex forward models. For relatively small numbers of iterations it was suggested that non-uniform error inflation for the sequence of assimilation cycles (leading to non-uniform update steps) could be beneficial for the results of the ES-MDA scheme (Evensen, 2018). We tested this idea in a small number of synthetic experiments (not shown), but did not find a striking improvement of the results. Still, we are of the opinion that related ideas to improve the tempering should be tested more extensively in future studies, for example by following the adaptive approach outlined by Iglesias and Yang (2021).

The PBS scheme is less affected by this problem, but a six-dimensional parameter space is already so large that the method cannot effectively represent the posterior distribution due to the curse of dimensionality that plagues importance sampling-based methods (Snyder et al., 2008). The PIES scheme presented here aims to overcome this issue by combining the ES-MDA scheme with the PBS scheme to take advantage of their individual strengths. In particular, the PIES scheme is (unlike the PBS) not confined to simply weighting the initial samples drawn from the prior. Instead a proposal distribution based on the ES-MDA is used to guide the attention of the importance sampling to areas of higher posterior probability mass. As such, we see that the PIES scheme markedly outperforms the PBS in terms of RMSE (cf. Table 1), but nonetheless still suffers from degeneracy. To further improve the PIES scheme, and potentially avoid degeneracy, future studies could explore the possibility of using more iterations in the ES-MDA that is used for the proposal distribution. An alternative path would be to explore iterative particle methods (Chopin and Papaspiliopoulos, 2020).

We have not used the gold standard technique for Bayesian inference, namely Markov chain Monte Carlo methods (e.g. MacKay, 2003; Gelman et al., 2013), because our likelihood evaluations are so expensive that the sequential exploration of the parameter space with tens or even hundreds of thousands of steps would not be possible in a realistic time frame. New data assimilation schemes, some of which are specifically designed to handle problems with expensive likelihoods (Garbuno-Inigo et al., 2020), could open new possibilities for drone flux measurements. Among these schemes, a particularly promising route could be to explore the adoption of machine learning-based emulators (Cleary et al., 2021) and active learning (e.g. Murphy, 2022) as steps to further improve the posterior estimates without considerably increasing the computational cost. The majority of this computational burden stems not primarily from the update steps themselves, but rather from the need to iteratively run an ensemble of LES. The cost of running a single LES with PALM given our experimental setup is on average in the order of 50 CPU hours. The cost of running PALM with a particular parameter combination varies considerably given the adaptive timestep in PALM, but this average cost gives an indication of the considerable computational effort involved. As such, the computational cost of the DA schemes can be measured directly in terms of the number of runs of LES ($N_r$) required to infer the posterior flux estimates. Herein, these fluxes are parameters rather than states, so we do not strictly need to run posterior predictions, thus lowering the computational costs. Still, the ES-MDA with $N_a = 2$ iterations and $N_e = 100$ ensemble members requires $N_r = N_a \times N_e = 200$ LES. The PIES scheme requires exactly the same number of LES as the ES-MDA. The non-iterative ES and PBS schemes, on the other hand, have a lower cost of $N_r = N_e = 100$ LES. Performing these DA

schemes together in the same experiment, i.e. with the same prior ensemble, has a lower cost than running them separately. In particular, while running the ES-MDA all the other schemes can effectively be run for free as benchmarks without the need for any additional LES. The total number of LES undertaken in this study was nonetheless considerable given that we performed 16 synthetic experiments and 18 real experiments, each with $N_r = 200$, amounting to a total of around 6800 LES. It is worth noting that this is still considerably less than the cost of a single Markov Chain Monte Carlo experiment, which typically requires in the order of $10^5$ model evaluations. Nonetheless, the cost of these simulations placed a considerable constraint on the number of experiments we could perform to explore an otherwise vast space of design choices that should be investigated in future studies.

## 5 Conclusion

To facilitate the development of drone flux measurements, a data assimilation framework for estimation of turbulent heat fluxes at the surfaces from sparse and noisy drone-based observations is presented using LES as a forward model. This framework allows explicit representation of the sequential collection of drone data, accounts for sensor noise, and includes uncertainty in boundary and initial conditions during the flux estimation. Different data assimilation schemes have been shown to markedly constrain the surface fluxes by using drone observations, with the ES-MDA scheme outperforming the three other tested schemes. Both synthetic and field experiments show promising results for homogeneous and stationary fluxes, which are in reasonable agreement with independent EC flux estimates. Increasing flight times, using observations from multiple drones, and narrowing the prior distributions of the initial conditions, are viable methods to further improve flux results. Sampling strategies prioritizing spatial exploration instead of temporal averaging at fixed positions enhance the non-linearities in the forward problem and can lead to biased flux results.

While the comparison here uses the simplifying assumptions of flux homogeneity, stationarity, and flat terrain, we emphasize that the drone data assimilation framework is not confined to these assumptions (as long as they can be accommodated in the forward model) and can thus readily be extended to more complex cases in future studies. Future effort could aim to apply this framework to estimate gas fluxes of e.g. $CO_2$ and methane, which would be another valuable contribution to Earth system science.

## Appendix A: Particle methods

Importance sampling lies at the core of particle (or sequential Monte Carlo) methods such as PIES and PBS. Rather than directly sampling from a target distribution of interest, this sampling method estimates expectations with respect to a target distribution through indirect Monte Carlo integration by drawing from a proposal (also known as importance) distribution that is easier to sample from (MacKay, 2003). In DA, and Bayesian inference more generally, the posterior is the target distribution of interest and the expectation of some function $g(\mathbf{x})$ with respect to the posterior is defined as (Särkkä, 2013)

$$\mathrm{E}\left[g(\mathbf{x})|\mathbf{y}\right] = \int g(\mathbf{x})p(\mathbf{x}|\mathbf{y})\,\mathrm{d}\mathbf{x}, \tag{A1}$$

where, for example, the expectation of $g(\mathbf{x}) = \mathbf{x}$ yields the posterior mean. Given $N_e$ independent samples from the posterior, $\mathbf{x}_i \sim p(\mathbf{x}|\mathbf{y})$, we could approximate the expectation in (A1) numerically using direct Monte Carlo integration through $\mathrm{E}[g(\mathbf{x})|\mathbf{y}] \simeq \frac{1}{N_e} \sum_{i=1}^{N_e} g\left(\mathbf{x}^{(i)}\right)$. Due to the law of large numbers and the central limit theorem, this approximation will converge almost surely to the true expectation as $N_e \to \infty$ with a standard error inversely proportional to $\sqrt{N_e}$ (Chopin and Papaspiliopoulos, 2020). In practice, it is rarely possible to generate independent samples directly from the posterior.

In importance sampling, we adopt a tractable proposal distribution $q(\mathbf{x})$ with (at least) the same support as the posterior. Multiplying the integrand with $1 = q(\mathbf{x})/q(\mathbf{x})$, (A1) can be expressed as

$$\mathrm{E}[g(\mathbf{x})|\mathbf{y}] = \int g(\mathbf{x}) \frac{p(\mathbf{x}|\mathbf{y})}{q(\mathbf{x})} q(\mathbf{x}) \,\mathrm{d}\mathbf{x} \simeq \frac{1}{N_e} \sum_{i=1}^{N_e} g(\mathbf{x}_i) \widehat{w}_i \,, \tag{A2}$$

where the scheme draws from $\mathbf{x}_i \sim q(\mathbf{x})$ so that the normalized weights can be defined as $\widehat{w}_i = p(\mathbf{x}_i|\mathbf{y})/q(\mathbf{x}_i)$. An obstacle remains in that we can only directly evaluate the un-normalized posterior $f(\mathbf{x}) = p(\mathbf{y}|\mathbf{x})p(\mathbf{x})$ and not the evidence $p(\mathbf{y})$ in the denominator of (3) because it is the integral of $f(\mathbf{x})$. Nonetheless, we can also approximate the evidence with importance sampling through $p(\mathbf{y}) = \int f(\mathbf{x}) \,\mathrm{d}\mathbf{x} \simeq \frac{1}{N_e} \sum_{i=1}^{N_e} \widetilde{w}_i$ where $\widetilde{w}_i = f(\mathbf{x}_i)/q(\mathbf{x}_i)$ are the *unnormalized*-weights. With this evidence approximation, we can now approximate (A1) as

$$\mathrm{E}[g(\mathbf{x})|\mathbf{y}] = \int \frac{g(\mathbf{x})f(\mathbf{x})}{q(\mathbf{x})p(\mathbf{y})} q(\mathbf{x}) \,\mathrm{d}\mathbf{x} \simeq \frac{1}{N_e} \sum_{i=1}^{N_e} g(\mathbf{x}_i) w_i \,, \tag{A3}$$

where the (*auto*-normalized) weights are given by $w_i = \widetilde{w}_i \left[\sum_{k=1}^{N_e} \widetilde{w}_k\right]^{-1}$ with the property the $\sum_{i=1}^{N_e} w_i = 1$. To ensure numerical stability, we use the log-sum-exp transformation when computing these weights (Murphy, 2022). The PBS scheme is obtained by using the prior as the proposal, i.e. $q(\mathbf{x}) = p(\mathbf{x})$, where (7) is for the particular case of a Gaussian prior and likelihood used herein. Similarly, the novel PIES scheme in Equation (8) is obtained by using the Gaussian distribution $\mathrm{N}(\widehat{\boldsymbol{\mu}}, \widehat{\boldsymbol{\mathcal{C}}})$ from the penultimate ES-MDA iteration as the proposal distribution.

As a final point, we emphasize that importance sampling results in a particle representation of the posterior through a sum of weighted Dirac delta functions centered on the sampled states $\mathbf{x}_i \sim q(\mathbf{x})$ of the form $p(\mathbf{x}|\mathbf{y}) \simeq \sum_{i=1}^{N_e} w_i \delta(\mathbf{x} - \mathbf{x}_i)$ (Särkkä, 2013). This point can be appreciated by recalling that the Dirac delta has the properties $\int \delta(\mathbf{x} - \mathbf{x}_i) \,\mathrm{d}\mathbf{x} = 1$ and $\int g(\mathbf{x})\delta(\mathbf{x} - \mathbf{x}_i) \,\mathrm{d}\mathbf{x} = g(\mathbf{x}_i)$, so that if we insert the particle representation in (A1) then

$$\mathrm{E}[g(\mathbf{x})|\mathbf{y}] \simeq \int \sum_{i=1}^{N_e} g(\mathbf{x}) w_i \delta(\mathbf{x} - \mathbf{x}_i) \,\mathrm{d}\mathbf{x} = \sum_{i=1}^{N_e} g(\mathbf{x}_i) w_i \,, \tag{A4}$$

which is the same as the result in (A3). Under this particle representation we can conceptualize the posterior distribution as a set of particles (or points) in parameter space whose probability mass are given by their weights.

## Appendix B: Ensemble Kalman methods

The ensemble Kalman filter (EnKF; Evensen, 1994) is a Monte Carlo version of the Kalman filter (Jazwinski, 1970; Särkkä, 2013). These schemes both make a Gaussian linear assumption, on top of the usual filtering assumptions of Markovian dynamics and conditionally independent observations. When these assumptions are satisfied, the exact filtering distribution is a Gaussian that is available analytically in closed form through the Kalman filtering equations. Unlike the original Kalman filter, the EnKF can still be used when these assumptions are violated. In fact, it is remarkably robust to such violations which explains why it is a widely used method for the typically nonlinear and high dimensional problems that arise in geoscientific data assimilation (Carrassi et al., 2018). These ensemble Kalman methods can also be applied to solving more general smoothing problems in which asynchronous observations are assimilated (Cosme et al., 2012). The ensemble smoother (ES; van Leeuwen and Evensen, 1996), a batch smoother version of the EnKF, and its iterative variants such as the ES-MDA (Emerick and Reynolds, 2013) have been shown to be particularly useful in the context of estimating static parameters in inverse problems (e.g. Evensen, 2018; Aalstad et al., 2018; Evensen, 2019; Garbuno-Inigo et al., 2020; Cleary et al., 2021; Alonso-González et al., 2022).

Here, the equations for both the stochastic ES and the ES-MDA are presented, while noting that a full derivation of the ensemble Kalman analysis equations can be found elsewhere (e.g. Evensen et al., 2022b). Let $N_a$ denote the number of assimilation cycles, then for the ES we set $N_a = 1$ while for the ES-MDA $N_a > 1$, typically with $N_a = 4$. The superscript $\ell$ indexes these iterations. Let $\mathbf{X}^{(\ell)} = \left[ \mathbf{x}_1^{(\ell)}, \ldots, \mathbf{x}_i^{(\ell)}, \ldots, \mathbf{x}_N^{(\ell)} \right]$ denote the $m \times N_e$ parameter matrix containing the ensemble ($i = 1, \ldots, N_e$) of parameter vectors $\mathbf{x}_i^{(\ell)}$ for iteration $\ell$. The subset of these parameters that are physically bounded have undergone the relevant analytic transformations for Gaussian anamorphosis (Bertino et al., 2003; Aalstad et al., 2018), and the corresponding inverse transforms are applied back to physical space when these are passed through the forward model $\mathcal{G}$. Similarly, let $\widehat{\mathbf{Y}}^{(\ell)} = \left[ \widehat{\mathbf{y}}_1^{(\ell)}, \ldots, \widehat{\mathbf{y}}_i^{(\ell)}, \ldots, \widehat{\mathbf{y}}_N^{(\ell)} \right]$ denote the predicted observation matrix containing the ensemble of predicted observations $\widehat{\mathbf{y}}_i^{(\ell)} = \mathcal{G}\left( \mathbf{x}_i^{(\ell)} \right)$. Then these stochastic ensemble Kalman methods proceed by initially drawing the initial parameters from the prior $\mathbf{x}^{(\ell=0)} \sim p(\mathbf{x})$, then for $\ell = 0 : (N_a - 1)$ iterations:

$$\mathbf{X}^{(\ell+1)} = \mathbf{X}^{(\ell)} + \mathbf{K}^{(\ell)} \left[ \mathbf{Y} - \left( \widehat{\mathbf{Y}}^{(\ell)} + \boldsymbol{\mathcal{E}}_\alpha^{(\ell)} \right) \right] , \tag{B1}$$

where $\mathbf{Y}$ is an $d \times N_e$ matrix with $N_e$ copies of the observation vector $\mathbf{y}$ while the observation error term is $\boldsymbol{\mathcal{E}}_\alpha^{(\ell)} = \sqrt{\alpha^{(\ell)}} \mathbf{R}^{1/2} \boldsymbol{\zeta}^{(\ell)}$ in which $\boldsymbol{\zeta}^{(\ell)}$ is an $d \times N_e$ matrix containing draws from a standard Gaussian N(0,1) and $\alpha^{(\ell)} = N_a$ is the observation error inflation coefficient. The so-called (ensemble) Kalman gain $\mathbf{K}^{(\ell)}$ is the $m \times d$ matrix

$$\mathbf{K}^{(\ell)} = \mathbf{C}_{\mathbf{X}\widehat{\mathbf{Y}}}^{(\ell)} \left( \mathbf{C}_{\widehat{\mathbf{Y}}\widehat{\mathbf{Y}}}^{(\ell)} + \alpha^{(\ell)} \mathbf{R} \right)^{-1} , \tag{B2}$$

where $\mathbf{C}_{\mathbf{X}\widehat{\mathbf{Y}}}^{(\ell)} = \frac{1}{N} \mathbf{X}^{(\ell)\prime} \widehat{\mathbf{Y}}^{(\ell)\prime\mathrm{T}}$ and $\mathbf{C}_{\widehat{\mathbf{Y}}\widehat{\mathbf{Y}}}^{(\ell)} = \frac{1}{N} \widehat{\mathbf{Y}}^{(\ell)\prime} \widehat{\mathbf{Y}}^{(\ell)\prime\mathrm{T}}$ are the $m \times d$ parameter-predicted observation covariance matrix and the $d \times d$ predicted observation covariance matrix, respectively, in which primes $(\cdot)'$ denote deviations from the ensemble mean.

*Code and data availability.* The code of the PALM model is freely available at palm.muk.uni-hannover.de. Drone measurements for the synthetic and real-world experiments together with the concurrent EC results, and the PALM steering file template are available under DOI https://doi.org/10.5281/zenodo.6769683.

*Author contributions.* Coneptualization: NP, KA, SW, MC, GK; Data curation: NP, AV; Formal analysis: NP, KA; Funding acquisition: NP, KA, SW, LMT, MC, GK; Writing - original draft preparation: NP, KA; Writing – review and editing: NP, KA, SW, LMT, MC, GK, AV, AvH.

*Competing interests.* The authors declare that they have no conflict of interest.

*Acknowledgements.* We thank Thomas Friborg for his valuable comments on our manuscript. We also thank Matthias Sühring for discussions and help with the PALM code. The work was supported by the Research Council of Norway (Projects #301552 "Upscaling hotspots - understanding the variability of critical land-atmosphere fluxes to strengthen climate models (Spot-on)" and #294948 "Terrestrial ecosystem-climate interactions of our EMERALD planet"). This work is a contribution to the strategic research initiative LATICE (Faculty of Mathematics and Natural Sciences, University of Oslo, Project #UiO/GEO103920) as well as the Centre for Computational and Data Science (dScience, UiO). Computations have been carried out on the high performance computer clusters of the Norwegian Research Infrastructure Services (Sigma2).

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
