# Peer review of "Inferring surface energy fluxes using drone data assimilation in large eddy simulations"

_Atmospheric Measurement Techniques, 2022_

## Author Comment (AC1)

We are grateful to the reviewer for the thoughtful comments and suggestions to our manuscript. We have compiled a revised version and in the following provide a point-by-point reply to all issues raised.

COMMENT # 1.1

*This review comes from a referee with a more mathematical/statistical background with very little practical experience from drone or tower measurements of the atmosphere.*

*The overall quality of this manuscript was very high. I believe it is well written and well structured. The mathematical framework is strong, and I appreciate the detail in describing the priors over the parameter distributions. I believe the use of data assimilation methods integrating drone data into LES is interesting and appropriate for the task at hand, and such approaches have a strong grounding of success in many other disciplines. The authors have compiled a comprehensive evaluation of several algorithms on a good synthetic baseline, and they have extended this work into a real-world setting for some qualitative conclusions. The algorithms were well documented and reasonably well explained, though a novel algorithm proposed in this paper feels quite unmotivated and underperforming. The outlook was well-considered and conclusions not over-stated, and the study opens doors into further experimental design questions for data collection, validation of eddy correction vs drone data assimilation, and algorithm selection for ensemble smoothing/inversion.*

**Reply:**

We would like to thank the reviewer for their positive and constructive comments. The specific comments related to the PIES scheme are addressed below in response to Comment #1.5.

COMMENT # 1.2

*Specific comments*

*The language and methods are that of operational forecasting for the formulation of the data assimilation problem. Yet the problem (1) is one distinctly of an inverse, "smoothing", problem. The authors are I believe aware of the new field arising using the language from Bayesian Inverse Problems and optimization, producing methods such as Ensemble Kalman Inversion (EKI); a method which encompasses both (ES) or (ES-MDA) by choice of different time-stepping schemes (e.g. ES-MDA is based on Bayesian tempering). Given the success of*

*this family of methods in the paper, I suggest the authors move their references from this field into the main body of the text from the appendix. I would also add a reference such as (Iglesias,Yang, 2021: https://doi.org/10.1088/1361-6420/abd29b) for tempering-based timesteps with EKI which may offer a new perspective on the methods.*

**Reply:**

We are indeed aware of this emerging field of ensemble Kalman inversion and are excited to see a broader unification of data assimilation and inverse modeling techniques under the common umbrella of Bayesian inference. In particular, we are adopting a broad definition of the term data assimilation from the Bayesian perspective outlined in (1). Under this definition, data assimilation is also concerned with parameter estimation in batch smoothing problems that arise in areas as diverse as reservoir history matching (2), snow reanalysis (3), and the present study. As the reviewer points out, this problem turns out to be identical to the Bayesian inverse problem formulation outlined by (4) that we cite in our manuscript. To make this connection clearer we have now also included more of the related studies (such as (5; 6)), as well as the study by Iglesias and Yang (2021) (7), to the reference list and moved these to earlier portions of the text. In particular, our broad definition of DA and the overlap with Bayesian inverse problems is now explicitly discussed towards the end of the Introduction and the link to ensemble Kalman inversion has been added to the section introducing the ES-MDA in Section 2.4:

**Changes:**

**2.1.4 Introduction**

...

This view implies that a mathematically optimal technique for consistent data-model fusion can be formulated as a kind of Bayesian inference problem (8; 10; 9; 11) , which is typically referred to as data assimilation (DA) or inverse modeling in the geosciences (12; 1). Herein, we adopt a broad Bayesian definition of the field of DA in line with (13). In addition to the conventional DA problem of state estimation, this definition also encompasses the problem of parameter estimation. The latter is often referred to as an inverse problem (4) rather than a DA problem. Since the flux estimation problem at hand is precisely such a parameter estimation or inverse problem we are also leaning on developments in this field (5; 6). In this study, we do not make any distinction between DA and inversion and take a unifying approach through the lens of Bayesian inference following (9). Such a unified view is especially helpful as the methods used herein can be applied in a hierarchical framework that jointly solve both state and parameter estimation problems (14).

...

**2.1.4 Data assimilation schemes**

...

At the root of these iterative schemes we find the idea of tempered transitions, which is a technique that is widely used in challenging Bayesian inference  tasks (15; 16; 7). This tempering, in combination with their derivative-free implementation, has placed iterative ensemble Kalman methods at the frontier of ongoing research in Bayesian inverse problems (4; 5; 6) which is helping to both formalize, improve, and generalize this family of methods (17; 7; 18; 19). The equations and workflow for the ES-MDA scheme used herein are presented in Appendix B.

COMMENT # 1.3

*It would be illustrative to unwrap the classic equation (1). For example the authors state L116 "G(·) is the forward model (e.g. RANS or LES)" but this is not generally true, it only contains RANS or LES.*

*In particular (1) hides the important presence of:*

*(i) observational map (here related to the experimental design of drone movements) and*

*(ii) the transformation map from "computational" Gaussian to "transformed parameters" to physical e.g. positive parameter distributions.*

*The inclusion of (i) could be used later to explicitly describe the the drone observations, such as the aggregation times vs timesteps and for the different experiments.*

*The inclusion of (ii) could be used in relation to the comment in L200-205 where it is mentioned that Kalman methods theory is based on Gaussian assumptions to explain why the parameters are defined to be transforms of Gaussians. It should not be forgotten that the theory of Kalman methods also relies on linearity, and such transformation introduces additional nonlinearity into the forward model. A comment here, along with an expansion of the forward map as "H ∘ F ∘ T= Observation_operator∘LES∘Transform" in (1) would illuminate this.*

**Reply:**

We have followed the reviewer's suggestion of unwrapping Equation (1) to make the various operations more explicit as outlined below.

**Changes:**

**2.1 Data assimilation framework**

...

The aim here is to infer surface fluxes of sensible and latent heat using sparse and uncertain drone measurements of meteorological variables in the atmospheric boundary layer. Solving this inverse problem requires a forward (or data generating) model that maps the parameters, namely the surface fluxes of interest and other uncertain boundary conditions, to the drone observations through

$$\mathbf{y} = \mathcal{G}(\mathbf{x}) + \boldsymbol{\epsilon}, \tag{1}$$

where $\mathbf{y} \in \mathbb{R}^d$ is the observation vector, $\mathcal{G}(\cdot)$ is the forward model, $\mathbf{x} \in \mathbb{R}^m$ is the target parameter vector, and $\boldsymbol{\epsilon} \in \mathbb{R}^d$ is the observation error.  In practice, $\mathcal{G}(\cdot)$ is a composition of multiple operations (c.f. 1)

$$\mathcal{G}(\mathbf{x}) = \mathcal{H}(\mathcal{M}(\mathcal{T}(\mathbf{x}))). \tag{2}$$

The inner operation, $\mathcal{T}(\cdot)$, is a transformation step that maps the parameters from an unbounded space to a bounded physical space. This step helps satisfy the Gaussian assumption of the ensemble Kalman methods while avoiding unphysical values (Section 2.1.2), although it adds an extra layer of non-linearity to the forward model. The subsequent middle operation, $\mathcal{M}(\cdot)$, is the dynamical model used to simulate the state of the boundary layer given the boundary conditions specified by the parameters. The outer operation, $\mathcal{H}(\cdot)$, is the observation operator that maps the states of the model to the corresponding predicted observations by extracting the flight paths of drones and (when necessary) performing temporal aggregation (see Section 2.1.3). By employing a turbulence-resolving LES  as opposed to a RANS model for the dynamics $\mathcal{M}(\cdot)$ in our forward model $\mathcal{G}(\cdot)$, we are able to gen-erate  the surface flux to drone observation mapping since the LES is run at an appropriate level of spatio-temporal detail.

Even in the absence of observation error,...

COMMENT # 1.4

> *The authors should present clear equations for the observational covariance matrices that arise from the different artificial experiments should be added. I am particularly interested in the apparent overfitting that occurs during the random sweeps in the synthetic experiment. For example, were matrices scaled by sqrt(T) (where T is the aggregation time difference) when moving to the shorter measurements in the random trajectories? More generally, was the shortest timescales for CLT approximation to provide a good estimate investigated (e.g. is aggregation of 10s enough to assume the effects of the nuisance is random)?*

**Reply:**

We have followed the reviewer's suggestion and presented clear equations for the observational error covariance matrices, including how we scaled these to account for the number of samples $S$ (what the reviewer calls $T$) in the averaging operations (factor $1/\sqrt{S}$ for error standard deviations, $1/S$ for error variances) and local mean gradients (introducing a factor $\sqrt{2}$ for error standard deviations, so $2$ for error variances). The use of Gaussian observation errors in this study is an assumption and, given CLT, it is more likely to hold for longer averaging periods (i.e. larger $S$). We did not test the appropriateness of the CLT approximation for shorter timescales per se, instead our experimental design was based on our prior expectations of the turbulence spectra and practical constraints related to the drones (battery time etc...). This could be an important topic to explore further in future work. We nonetheless added changes to Section 2.1.3 to clarify some of these concerns as outlined below

**Changes:**

**2.1.3 Drone measurements, observations and errors**

. . .

Systematic errors that occur for error distributions that are asymmetrically distributed with respect to zero, are assumed to be negligible. This leads to the following definition for the diagonal observation error covariance matrix $\mathbf{R} \in \mathbb{R}^{d \times d}$ employed in this study

$$\mathbf{R} = \mathrm{diag}\left(\boldsymbol{\tau} \odot \boldsymbol{\sigma}^2\right), \tag{5}$$

where $\mathrm{diag}(\cdot)$ is the diagonal operator that converts a vector to a diagonal matrix, $\boldsymbol{\tau} \in \mathbb{R}^d$ is a scaling vector, $\odot$ denotes the element-wise product, $\boldsymbol{\sigma} \in \mathbb{R}^d$ contains the measurement error standard deviation for each observation. The elements of the scaling vector are defined as follows

$$\tau_i = \begin{cases} 1/S & \text{if mean,} \\ 2/S & \text{if local mean gradient,} \end{cases} \tag{6}$$

where $S$ is the number of measurement samples that are averaged to form an observation. As elaborated in Section 2.2, we test two types of flight plans. The first type involves step-wise vertical profiles while the drones hover in place for a 2 minute averaging period with a 10 s sampling interval such that $S = 12$. The second type involves random exploration where no averaging is performed such that $S = 1$. In summary, following independent Gaussian error propagation, this observation error covariance matrix implies that observation errors are uncorrelated, decrease with number of samples $S$ in an averaging period, and are larger for local mean gradients than for

The elements of σ are determined by the measurement error standard deviation of the respective sensors. For temperature measurements on drones,...

COMMENT # 1.5

*The novel PIES algorithm unfortunately does not seem to be effective, given that PIES has suffered similar collapse to the PBS in synthetic experiments. Therefore the discussion of its performance with KLD (e.g. in L413) or RMSE should probably be cautious at best as clearly it is stuck in a suboptimal local minimum. Discussion of actual performance indicators should be limited to the Kalman methods, ES and ES-MDA, that appear to have at least retained posterior spread around the truth. I feel that there is not enough motivation as to why the PIES algorithm was developed, what theory or heuristics lead the authors to believe that it should work, and whether it performed as expected in practice. I think it's relevant considering the other works are available for overcoming this degeneracy e.g. L271: "several more sophisticated variants are shown to have potential to overcome this (van Leeuwen et al. 2019)".*

**Reply:**

The PIES scheme did not turn out to be particularly effective in this study and suffered from the same degeneracy as the simpler PBS scheme. We suspect that this could be improved by running more iterations of the ES-MDA and using the final (rather than pen-ultimate) iteration as the posterior, but that would come at a considerable increase in computational cost. Following the reviewer's suggestion, we have thus removed the discussion about the KLD of the PIES scheme on L413. We have nonetheless retained the discussion of the RMSE of the particle methods since this is a point metric based on the ensemble mean (rather than the entire ensemble) and we have in the same section (Section 3.2) made it clear that these schemes were degenerate with effective sample sizes of 1. A discussion of the motivation for testing the PIES method in this paper has also been added. It is worth emphasizing that all the non-iterative schemes (ES, PBS, PIES) can essentially be run for free (other than the cost of the assimilation step) while running the ES-MDA iterations. In particular, they do not require any additional LES runs. As such, including these schemes as tests or benchmarks for the performance of the ES-MDA does not add any noticeable computational burden to our experiments.

**Changes:**

**2.1.4 Data assimilation schemes**

. . .

The motivation for pursuing the PIES scheme is that the ES-MDA produces a biased approximation of the posterior for non-linear forward models (16). Although this bias is typically less severe than that of non-iterative ensemble Kalman methods (2), it would nonetheless be advantageous to find efficient methods to reduce it. PIES is a straightforward translation of the scheme of (20) to iterative ensemble smoothers such as the ES-MDA. As such, PIES can be viewed as a simple extension of the ES-MDA that does not necessarily impose any noticeable computational burden and might improve performance. As with all particle methods, the effective sample size can be used to diagnose degeneracy in the ensemble of particles (21). A low ($\ll N_e$) effective sample size indicates degeneracy due to the fact that the proposal is too far from the target posterior. . . .

**3.1 Synthetic experiments**

. . .

The ES-MDA  scheme gives the largest information gain from the prior to posterior, as indicated by  its KLD.

COMMENT # 1.6

*I think more should be discussed in moving the algorithm application from synthetic data to field data, (alongside the comparison of data to EC). Is it possible to obtain a plot of the parameter prior and marginal posteriors for the ES-MDA for (e.g. repeating Figure 3 for the field data). Is there anything to suggest that significant structural model errors appear (as compared with the synthetic data) or are they captured well by the pairing of the LES model and choice of observational covariances in the inverse problem.*

**Reply:**

We agree that structural model errors should be clarified when moving the algorithm from synthetic to field data, so we suggest to add the text below to Section 2.1. Overall, we believe that the main structural errors for field experiments are due to topography and spatio-temporal flux variability. Assuming flat terrain, as well as homogeneous and stationary surface fluxes are simplifications of reality, which should be improved in future studies.

As requested by the reviewer, Figure R1 below shows an example of the marginal distributions for a field data experiment (the same flight as used in Figure 5 of the main article). Note the important difference to the synthetic experiments that true

parameter values are not known (vertical dashed lines for H and LE only show independent EC estimates). We do not see indications of significant structural model errors from these distributions, indicating that the choice of LES model and observation operator give an appropriate representation of reality.

**Changes:**

**2.1 Data assimilation framework**

. . .

In the real experiments, where we compare with independent EC data, some of the mismatch between the EC estimates and drone-based inferences will undoubtedly be due to the strong assumptions made in the respective approaches. Given the level of realism in LES, these structural model errors introduced when moving the algorithm application from synthetic to field data are likely dominated by simplifications of topography and spatio-temporal flux variability. The Bayesian approach to inference

. . .

COMMENT # 1.7

*Technical corrections*

*L109 "we do not argue that this comparison offers validation per se - only a plausibility check". Can the authors instead write what they wished to see/gain from the experiment?*

**Reply:**

As the two methods estimate the surface fluxes over slightly different footprint areas, we wish to see that they agree in the estimated order of magnitude of fluxes and the relative flux variability. We agree with the reviewer that this understanding of "plausibility check" should be clarified.

**Changes:**

**Introduction**

. . .

To be clear, given the differences in footprint and underlying assumptions, we do not argue that this comparison offers a validation per se – only a plausibility check of the estimated order of magnitude of fluxes and their relative variability.

COMMENT # 1.8

[Figure]

Figure R1: Marginal parameter distributions for the prior (black) as well as the ES-MDA (with $N_a = 2$ iterations, red), ES (blue), PBS (yellow shading shows the central $95^{th}$-percentile range), and PIES (green shading shows the central $95^{th}$-percentile range) posterior estimates along with the location of the EC flux estimates (black dashed vertical line) for flight 1 of the Iškoras campaign, a step profile on 2020-07-27 with takeoff at 15:20 local time.

> *Throughout, single or double quotes appear backward before quotations, typically from using character ' and not ' in latex*

**Reply:**

Fixed, thanks for spotting this.

COMMENT # 1.9

*The authors describe all parameters that are not "H" or "LE" as nuisance parameters, but then still proceed to learn them. Perhaps I am mistaken, but I thought that nuisance parameters are not learnt in DA - rather they are parameters whose effect is considered to add additional noise in the observation functional in place of trying to learn them in the scheme, I would say their description is underplaying the work that they subsequently undertake*

**Reply:**

Indeed, our interest is in H and LE, but the other parameters are learned from the data and then 'integrated out' by focusing on the marginal distribution of of H and LE. We were of the impression that our use of the word nuisance was in line with the norm in Bayesian statistics following (10) and (22) as well as (11) which is a standard reference in the field. This procedure may not be the norm in conventional state estimation-based DA, but we have now made it clear (see response to Comment # 1.2) that we are adopting a broader Bayesian definition of the term that includes parameter estimation (and marginalization). We have clarified this understanding of 'nuisance' parameters in the following change of the manuscript.

**Changes:**

**2.1.1 LES model and parameters**

. . .

Of these six parameters, the primary interest is in H and LE while the remaining four parameters can be regarded as 'nuisance' parameters (10)(22; 10; 11). The nuisance parameters are still inferred from the data, but are then implicitly 'integrated out' as we primarily focus on the marginal posterior distributions of H and LE. . . .

COMMENT # 1.10

*Presentation of Table 1 naturally should be alongside that of Figure 3 as they are both inter-algorithm performance comparison. Figure 4 should come after this as it has already selected the "best" algorithm and addresses a different scientific question*

**Reply:**

We agree that this change of order can clarify the manuscript and are happy to follow the reviewer's suggestion. To emphasize that the table presents average statistics over a number of synthetic experiments, we also suggest to add the following sentence to the paragraph describing the table.

**Changes:**

**3.1 Synthetic experiments**

. . .

Varying the sampling strategies (step profile vs random exploration), flight time (12  vs 24 minutes), number of drones (1 vs 5), uncertainty in initial conditions (narrow vs broad), and the geostrophic wind speed (1.5 vs 6.0 m s$^{-1}$) led to a total of 16 synthetic experiments.

COMMENT # 1.11

*Figure 4 mention the spread of the violin plots (95%) in the caption*

**Reply:**

The violins are plotted with Matplotlib Violinplot, where the caps mark the extrema of the ensemble. We propose to add the following sentence to the caption of Figure 4 to clarify this.

**Changes:**

**Figure 4 caption:**
The caps of the violins mark the extrema of the ensemble and the dots the mean values.

COMMENT # 1.12

*L490 typo "constrains"*

**Reply:**

Fixed, thanks for spotting this typo.

COMMENT # 1.13

*L388 - either should say "see discussion below" or "see Section 4" depending on what it refers to. (likewise L460 could just read "see Section 4.2").*

**Reply:**

We have now referred to the appropriate part of the manuscript, namely Section 4.2. Moreover, we have added a sentence to provide an example of what kind of external information we are referring to.

**Changes:**

**4.2 Possible improvements**

...

A complementary approach could also be to directly incorporate land cover information, e.g. from satellite retrievals (23), into the design of flux maps in the turbulence simulation as was done in (24).

COMMENT # 1.14

*Were more than two inflation steps tried with ES-MDA?*

**Reply:**

In the pilot phase of this study, we did try a few experiment with more iterations, but quickly realized, that given our allocation of computational resources we could not afford to test this systematically across many different experiments in our study. We chose to prioritize more experiments (including field data) with fewer iterations. So we can only hypothesize that more iteration in the ES-MDA scheme would give an improved performance (which should be addressed in future studies). In this context, we should also explore reducing the ensemble size and increasing the number iterations (while keeping the number of simulations fixed), to identify the optimal ratio of this computational trade-off. The paragraph we added to Section 4.3 (in relation to Comment #2.2 by Reviewer 2) summarizes these considerations.

COMMENT # 1.15

*Figure 5 - mention these are posterior draws of ES-MDA on the caption.*

**Reply:**

We have followed the reviewer's suggestion as shown below.

**Changes:**

**Figure 5 caption:**
Drone observations and posterior ensemble predictions from the ES-MDA for flight 1 of the Iškoras campaign, a step profile on 2020-07-27 with takeoff at 15:20 local time. The upper panels show the successive 2-min mean values, whereas the lower panels show the local mean gradients. The line colors of the vertical profiles for the  $N_e = 100$ posterior ensemble members from the ES-MDA correspond to their log-likelihood with more likely values in yellow and less likely values in blue. The prior

predictions are not shown, because their range is so wide that one could not see any details in the posterior profiles.

COMMENT # 1.16

*L455 "...compared to the less calibrated uncertainty estimates of the EC technique" - I'm not sure what this means here. Please rephrase*

**Reply:**

We agree that this can be clarified and suggest the following changes.

**Changes:**

**2.3 Field experiments**
. . .
Along with the EC fluxes, EddyPro also  estimates their random error (the variance of the flux covariance) due to sampling errors that arise from the small number of large eddies that dominate the flux during typical sampling periods following (25).

**4.1 Potential and limitations of drone data assimilation**
Using typical sensor configuration and flight times of small drones, we find a relatively large posterior spread of the surface fluxes, compared to the  uncertainty estimates of the EC technique that are solely accounting for sampling errors arising from the small number of large eddies captured in the 30-min flux interval. . .

COMMENT # 1.17

*Figure 6 - the uncertainties for EC are very small in Figure 6 (in relation to drone measurements), and grow with the value of the flux. Is this explainable? If so, is it unusual that the drone measurement uncertainty does not appear to depend on this?*

**Reply:**

We thank the reviewer for this interesting observation. Drone-DA uncertainty estimates are largely a result of our experimental design (flight time, sensor noise, etc.) and the prior distributions we used. These settings were kept constant in our field experiments shown in Figure 6, which would explain why the uncertainty estimates are largely constant. For EC, the used method by (25) to estimate the relative error of

the flux through the variance of the flux covariance, i.e. the sampling uncertainty. As shown in the original paper of the method (25), the relative error is largely constant (around 10-30%) over a range of flux magnitudes, wind speeds, and even ecosystem types (i.e. forested vs agricultural surfaces). It is therefore indeed expected that the absolute random error increases linearly with flux magnitude, as also seen in Figure 6 of our study. In sum, the error bars shown in Figure 6 measure fundamentally different uncertainties (epistemic for the drone DA, vs aleatoric for EC). We propose to add the following sentence describing this observation in the discussion.

**Changes:**

**3.2 Field experiments**
As the drone-DA uncertainty estimates are largely a result of our experimental design (flight time, sensor noise, etc.) and the used prior distributions, all 18 flights show largely the same epistemic uncertainty. For the EC estimates, error bars in Figure 6 only indicate the absolute aleatoric uncertainty due to sampling limitations, which is expected to increase with flux magnitude (25).

COMMENT # 1.18

*L304 "penultimate iteration of ES-MDA... in practice it may be better to use the posterior from the final iteration". Perhaps state precisely what the algorithm should use, then afterward mention what approximation is made for computational considerations.*

**Reply:**

Done.

**Changes:**

**2.1.4 Data assimilation scheme**
. . .
 Importance sampling is more effective the closer the proposal is to the target posterior distribution (8). So in theory it would be better to use the posterior estimate from the final (rather than penultimate) iteration of the ES-MDA for the proposal in PIES, but this would come at a high computational cost of requiring an additional round of runs of the LES ensemble.

COMMENT # 1.19

*L477 - more detail in the list of improvements. E.g more assimilation cycles should improve nonlinearity, more ensembles will improve the monte-carlo approximation, Gaussian processes could be used for increasing the smoothness of the cost landscape.*

**Reply:**

These are nice suggestions. We propose to incorporate them with the following changes in the sentence.

**Changes:**

**4.2 Possible improvements**

 Improved surface flux inferences can be achieved in a number of different ways, including technical improvements during data collection, such as using higher quality sensors, more drones, and better quantifying initial and boundary conditions, as well as modifying the data assimilation framework by using a larger ensemble size to improve the Monte Carlo approximation, more assimilation cycles  to better account for nonlinearity, emulators to increase the smoothness of the likelihood function (18), and a higher spatial resolution of the LES model to reduce structural model errors.

COMMENT # 1.20

*L462 - This is outside of my domain knowledge. But are there any high-level references that could be provided towards the present state and future progression to address the "engineering and legal challenges" of using drones to collect data?*

**Reply:**

The legal framework for drone applications is very country-specific and cannot be readily forecast. Typically, the legislation allows for the common use cases for drones (below 120 m above ground level and within visual line of sight) and describes possibilities to acquire permits for more advanced use cases. We understand that a reference to some legal guidelines is needed to clarify this aspect, so we propose to add a reference to an article focusing on a discussion of the European airspace regulations.

**Changes:**

**4.1 Potential and limitations of drone data assimilation**

. . .

Most applications of drones are currently still restricted to manual flights with a human pilot in charge of the system [see, e.g., [26] for a discussion of European airspace regulations].

**REFERENCES**

[1] G. Evensen, F. C. Vossepoel, and P. J. van Leeuwen, *Data Assimilation Fundamentals: A Unified Formulation of the State and Parameter Estimation Problem*. Springer Textbooks in Earth Sciences, Geography and Environment, Cham: Springer International Publishing, 2022.

[2] A. A. Emerick and A. C. Reynolds, "Ensemble smoother with multiple data assimilation," *Computers & Geosciences*, vol. 55, pp. 3–15, June 2013.

[3] K. Aalstad, S. Westermann, T. V. Schuler, J. Boike, and L. Bertino, "Ensemble-based assimilation of fractional snow-covered area satellite retrievals to estimate the snow distribution at Arctic sites," *The Cryosphere*, vol. 12, p. 247–270, 2018.

[4] A. M. Stuart, "Inverse problems: A Bayesian perspective," *Acta Numerica*, vol. 19, pp. 451–559, May 2010.

[5] M. A. Iglesias, J. H. Law, and A. M. Stuart, "Ensemble Kalman methods for inverse problems," *Inverse Problems*, vol. 29, no. 4, p. 045001, 2013.

[6] C. Schillings and A. M. Stuart, "Analysis of the Ensemble Kalman Filter for Inverse Problems," *SIAM Journal on Numerical Analysis*, vol. 55, no. 3, pp. 1264–1290, 2017.

[7] M. Iglesias and Y. Yang, "Adaptive regularisation for ensemble kalman inversion," *Inverse Problems*, vol. 37, no. 2, p. 025008, 2021.

[8] D. J. C. MacKay, *Information Theory, Inference, and Learning Algorithms*. Cambridge University Press, 2003.

[9] S. Särkkä, *Bayesian Filtering and Smoothing*. Cambridge University Press, 2013.

[10] E. Jaynes, *Probability Theory: The Logic of Science*. Cambridge University Press, 2003. doi:10.1017/CBO9780511790423.

[11] A. Gelman, J. Carlin, H. Stern, D. Dunson, A. Vehtari, and D. Rubin, *Bayesian Data Analysis*. Chapman and Hall/CRC, 3 ed., 2013.

[12] A. Carrassi, M. Bocquet, L. Bertino, and G. Evensen, "Data assimilation in the geosciences: An overview of methods, issues, and perspectives," *WIREs Climate Change*, vol. 9, Sept. 2018.

[13] G. Evensen, F. C. Vossepoel, and P. J. van Leeuwen, *Data Assimilation Fundamentals*. Springer, 2022.

[14] M. Katzfuss, R. S. Stroud, and C. K. Wikle, "Ensemble Kalman Methods for High-Dimensional Hierarchical Dynamic Space-Time Models," *Journal of the American Statistical Association*, vol. 115, no. 530, pp. 866–885, 2020.

[15] R. M. Neal, "Sampling from multimodal distributions using tempered transitions," *Statistics and Computing*, vol. 6, pp. 353–366, Dec. 1996.

[16] A. S. Stordal and A. H. Elsheikh, "Iterative ensemble smoothers in the annealed importance sampling framework," *Advances in Water Resources*, vol. 86, pp. 231–239, Dec. 2015.

[17] A. Garbuno-Inigo, F. Hoffmann, W. Li, and A. M. Stuart, "Interacting langevin diffusions: Gradient structure and ensemble kalman sampler," *SIAM Journal on Applied Dynamical Systems*, vol. 19, pp. 412–441, Jan. 2020.

[18] E. Cleary, A. Garbuno-Inigo, S. Lan, T. Schneider, and A. M. Stuart, "Calibrate, emulate, sample," *Journal of Computational Physics*, vol. 424, p. 109716, Jan. 2021.

[19] O. Dunbar, A. Duncan, A. Stuart, and M.-T. Wolfram, "Ensemble Inference Methods for Models With Noisy and Expensive Likelihoods," *SIAM Journal on Applied Dynamical Systems*, vol. 21, no. 2, pp. 1539–1572, 2022.

[20] N. Papadakis, E. Mémin, A. Cuzol, and N. Gengembre, "Data assimilation with the weighted ensemble Kalman filter," *Tellus A: Dynamic Meteorology and Oceanography*, vol. 62, pp. 673–697, Jan. 2010.

[21] N. Chopin and O. Papaspiliopoulos, *An Introduction to Sequential Monte Carlo*. Springer, 2020.

[22] G. Bretthorst, *Bayesian Spectrum Analysis and Parameter Estimation*. Springer, 1988.

[23] K. Aalstad, S. Westermann, and L. Bertino, "Evaluating satellite retrieved fractional snow-covered area at a high-Arctic site using terrestrial photography," *Remote Sensing of Environment*, vol. 239, p. 111618, 2020.

[24] L. D. van der Valk, A. J. Teuling, L. Girod, N. Pirk, R. Stoffer, and C. C. van Heerwaarden, "Understanding wind-driven melt of patchy snow cover," *The Cryosphere*, 2022.

[25] P. L. Finkelstein and P. F. Sims, "Sampling error in eddy correlation flux measurements," *Journal of Geophysical Research: Atmospheres*, vol. 106, pp. 3503–3509, Feb. 2001.

[26] E. Bassi, "From Here to 2023: Civil Drones Operations and the Setting of New Legal Rules for the European Single Sky," *Journal of Intelligent & Robotic Systems*, vol. 100, pp. 493–503, Nov. 2020.

[27] R. T. Palomaki, N. T. Rose, M. van den Bossche, T. J. Sherman, and S. F. J. De Wekker, "Wind Estimation in the Lower Atmosphere Using Multirotor Aircraft," *Journal of Atmospheric and Oceanic Technology*, vol. 34, pp. 1183–1191, May 2017.

---

## Author Comment (AC2)

We are grateful to the reviewer for the thoughtful comments and suggestions to our manuscript. We have compiled a revised version and in the following provide a point-by-point reply to all issues raised.

COMMENT # 2.1

*My background is mostly in data assimilation but I am little familiar with surface energy fluxes.*

*The manuscript by Pirk and co-authors introduces existing data assimilation methodology (plus a new assimilation method hybridising two previous methods) into a new application area of surface energy fluxes observations by new autonomous technology (drones). The topic is interesting and has practical outcomes for the best use of drones and the further exploitation of a promising technology.*

*The manuscript is very nicely written and is at a very mature stage already, making an enjoyable read. The methods are well presented, evaluated rigorously and the results make a convincing case to take the methodology forward. I only have a few minor questions.*

**Reply:**

Thanks for the nice comments!

COMMENT # 2.2

*On the data assimilation side I appreciate the introduction of the PIES scheme, which is original to my knowledge. The PIES scheme does not seem to bring much improvement and the authors are open on the shortcomings of the method. What I am missing is a sentence explaining the reasoning behind the PIES scheme: why replace the penultimate iteration of the ES-MDA method and not other ones? Otherwise the comparison of the assimilation methods is done in a correct way. An indication of their respective computational costs would be useful as a perspective.*

**Reply:**

We followed the reviewer's suggestion, which is related to Comment #1.5 by Reviewer 1, and expanded on the motivation and reasoning behind the PIES scheme as well as the computational costs of the respective DA schemes used in our study. To clarify the notation, given that we now use the symbols N (normal distribution) $N_a$

(number of assimilation cycles) and $N_r$ (number of LES runs in an experiment), we have changed the symbol for number of ensemble members from $N$ to $N_e$ throughout the manuscript.

**Changes:**

**2.1.4 Data assimilation schemes**

...

 Importance sampling is more effective the closer the proposal is to the target posterior distribution (8). So in theory it would be better to use the posterior estimate from the final (rather than penultimate) iteration of the ES-MDA for the proposal in PIES, but this would come at a high computational cost of requiring an additional round of runs of the LES ensemble. The motivation for pursuing the PIES scheme is that the ES-MDA produces a biased approximation of the posterior for non-linear forward models (16). Although this bias is typically less severe than that of non-iterative ensemble Kalman methods (2), it would nonetheless be advantageous to find efficient methods to reduce it. PIES is a straightforward translation of the scheme of (20) to iterative ensemble smoothers such as the ES-MDA. As such, PIES can be viewed as a simple extension of the ES-MDA that does not necessarily impose any noticeable computational burden and might improve performance. As with all particle methods, the effective sample size can be used to diagnose degeneracy in the ensemble of particles (21). A low ($\ll N_e$) effective sample size indicates degeneracy due to the fact that the proposal is too far from the target posterior. ...

**4.3 Data assimilation schemes for turbulent transport**

...

The majority of this computational burden stems not primarily from the update steps themselves, but rather from the need to iteratively run an ensemble of LES. The cost of running a single LES with PALM given our experimental setup is on average in the order of 50 CPU hours. The cost of running PALM with a particular parameter combination varies considerably given the adaptive timestep in PALM, but this average cost gives an indication of the considerable computational effort involved. As such, the computational cost of the DA schemes can be measured directly in terms of the number of runs of LES ($N_r$) required to infer the posterior flux estimates. Herein, these fluxes are parameters rather than states, so we do not strictly need to run posterior predictions, thus lowering the computational costs. Still, the ES-MDA with $N_a = 2$ iterations and $N_e = 100$ ensemble members requires $N_r = N_a \times N_e = 200$ LES. The PIES scheme requires exactly the same number of LES

as the ES-MDA. The non-iterative ES and PBS schemes, on the other hand, have a lower cost of $N_r = N_e = 100$ LES. Performing these DA schemes together in the same experiment, i.e. with the same prior ensemble, has a lower cost than running them separately. In particular, while running the ES-MDA all the other schemes can effectively be run for free as benchmarks without the need for any additional LES. The total number of LES undertaken in this study was nonetheless considerable given that we performed 16 synthetic experiments and 18 real experiments, each with $N_r = 200$, amounting to a total of around 6800 LES. It is worth noting that this is still considerably less than the cost of a single Markov Chain Monte Carlo experiment, which typically requires in the order of $10^5$ model evaluations. Nonetheless, the cost of these simulations placed a considerable constraint on the number of experiments we could perform to explore an otherwise vast space of design choices that should be investigated in future studies.

COMMENT # 2.3

*The synthetic experiments results seem to argue against the random exploration flight strategy, although for a reason related to the data assimilation technique (their effective observation errors are smaller). The authors should insist that their experiments do not disqualify the random flight strategy but may want to devise their observation representation errors more carefully.*

**Reply:**

We completely agree with this point, which is related to comment #1.4 by Reviewer 1, and clarified that our random exploration strategy used no temporal averaging in all relevant sentences of the manuscript.

**Changes:**

**Abstract**
...
Sampling strategies prioritizing space-time exploration  without temporal averaging, instead of hovering at fixed locations while averaging, enhance the non-linearities in the forward model and can lead to biased flux results with ensemble-based assimilation schemes.

**2.1.3 Drone measurements, observations and errors** (in the added paragraph about the error covariance matrix in relation to comment #1.4 by Reviewer 1)
...

The second type involves random exploration where no averaging is performed such that $S = 1$.

**4.2 Possible improvements**

...

The results indicate that both methods can constrain the surface fluxes, but random exploration without averaging multiple measurements for an observation can give biased flux results. These biases are likely due to shortcomings of the assimilation schemes used when dealing with strongly non-linear forward models rather than the sampling strategy itself, and so could be alleviated by improving the assimilation algorithms.

COMMENT # 2.4

*There is only one difference between the synthetic case and the real observations case and that is the independent versus correlated H and LE parameters. The authors do not come back to this difference in the discussions: does the correlation of parameters work well or should it be done differently?*

**Reply:**

We agree that the effect of prior parameter correlations should be brought up again in the discussion. We propose to add the following sentences to Section 4.2 (Possible improvements).

**Changes:**

**4.2 Possible improvements**

...

Our framework allows to add further information to the priors through correlations between individual parameters, which we only used for H and LE in our field experiments. The effect of these prior parameter correlations was mostly a slightly more effective exploration of the parameter space, but future studies could investigate how this feature can be used to reduce the computational costs with expensive models like LES.

COMMENT # 2.5

*The authors also use several statistical metrics to evaluate the methods from the classical RMSE and bias to the CRPS and KLD. It would be interesting to have the authors recommendation on how useful or redundant these metrics are in practice.*

**Reply:**

In our view, the presented metrics are all useful, as they quantify different aspects of the parameter distributions. RMSE and bias are standard metrics, typically used to compare the fit of point estimates. We also use the less known CRPS to compare the fit of the entire ensemble to the known true values. In that sense, CRPS captures a similar property as RMSE and bias, but we still think it's valuable to report all of them. Lastly, KLD compares the posterior and prior, and can be thought of as a measure of information gain, which is different from the fit of the data. In sum, we would recommend to report all four metrics in studies like ours.

**Changes:**

**2.2 Synthetic experiments**

...

These four metrics quantify different aspects of the fit and information gain of parameter distributions and can hence give a more holistic evaluation of a synthetic experiment.

COMMENT # 2.6

*Detailed comments and typos:*

*-l.15: "variance" is missing "minimum variance".*

**Reply:**

Technically, the degenerate posteriors of the PBS and PIES schemes have the minimum variance, but this is not desirable in this case. So we propose to clarify this sentence using "well-calibrated posterior uncertainty" instead of "low bias and variance".

**Changes:**

**Abstract**

...

It is shown that an iterative ensemble smoother outperforms both the non-iterative ensemble smoother and the particle batch smoother in the given problem, yielding  well-calibrated posterior uncertainty with continuous ranked probability scores of 12 W m$^{-2}$ for both H and LE with standard deviations of 37 W m$^{-2}$ (H) and 46 W m$^{-2}$ (LE) for a 12 min vertical step profile by a single drone.

SMALL CAPS: COMMENT # 2.7

*- l.239 "in a cyclic manner": do you mean the model domain is cyclic?*

**Reply:**

No, in this case we mean that the local mean gradients, i.e. the difference between the mean values at two (vertical) measurement levels, are also calculated for the first measurement location, but using the difference to the last measurement level. We propose to clarify this with the following change in the sentence.

**Changes:**

**2.1.3 Drone measurements, observations and errors**

. . .
This is done in a cyclic manner through the measurement locations, so that the local gradient at the first position is calculated as the difference to the last location.

COMMENT # 2.8

*- l.245: I miss an argument that the temporal representativity is somehow related to the spatial representativity of the observations, the discrepancy between the size of the instrument on the drone and the LES cell dimension.*

**Reply:**

We agree that the spatio-temporal aspect of representativeness errors between observations and model should be clarified. Key to this issue is the rotor wash from the drone that mixes the air around the drone, making its measurements less localized – and more representative for spatial scales similar to our LES grid spacing. We propose to add the following sentence to the paragraph in Section 2.1.3.

**Changes:**

**2.1.3 Drone measurements, observations and errors**

. . .
The related spatio-temporal representativeness errors are affected by the rotor wash from the drone that mixes the air around the drone and makes its measurements more representative for spatial scales similar to the LES grid spacing.

COMMENT # 2.9

*- l.255: Can you explain why 2 m/s errors on wind speed is "conservative"?*

**Reply:**

The studies we refer to for estimation of horizontal wind speeds from Inertial Measurement Unit data of multi-copter drones (e.g., Palomaki et al., 2017) report measurement uncertainties of less than 0.5 m/s. Since we did not evaluate this uncertainty for our drones, we decided to use a larger value of 2.0 m/s, to avoid underestimating this uncertainty.

**Changes:**

**2.1.3 Drone measurements, observations and errors**
. . .
For the horizontal wind speed $U$, the standard deviation for the measurement error is conservatively estimated to be $2.0 \, \mathrm{m\,s^{-1}}$. Other studies using Inertial Measurement Unit data of multi-copter drones for wind estimation report measurement uncertainties of less than $0.5 \, \mathrm{m\,s^{-1}}$ (27), but since we did not evaluate this uncertainty for our drones, we decided to use a somewhat larger value to avoid underestimating this uncertainty.

COMMENT # 2.10

*- l.283: "the" is missing before EnKF.*

**Reply:**

Fixed, thanks.

COMMENT # 2.11

*- l.420: "mean local differences": this notion is maybe familiar to the surface flux community but I would needed a little definition (horizontal gradient? Positive in which direction?)*

**Reply:**

We apologize for this confusion, which is caused by inconsistent semantics in this case. We meant to refer to the "local mean gradients" as defined in the manuscript,

but wrote "mean local differences".

**Changes:**

**3.2 Field experiments**

. . .

The measured mean values and  local mean gradients are generally well reproduced by the posterior LES ensemble.

COMMENT # 2.12

*- l.616 and 617: "an $d \times N$ matrix" should sound better as "a $d \times N$ matrix".*

**Reply:**

Since "m" in "matrix" is a consonant, the use of "a" over "an" should be correct.

**References**

[1] G. Evensen, F. C. Vossepoel, and P. J. van Leeuwen, *Data Assimilation Fundamentals: A Unified Formulation of the State and Parameter Estimation Problem*. Springer Textbooks in Earth Sciences, Geography and Environment, Cham: Springer International Publishing, 2022.

[2] A. A. Emerick and A. C. Reynolds, "Ensemble smoother with multiple data assimilation," *Computers & Geosciences*, vol. 55, pp. 3–15, June 2013.

[3] K. Aalstad, S. Westermann, T. V. Schuler, J. Boike, and L. Bertino, "Ensemble-based assimilation of fractional snow-covered area satellite retrievals to estimate the snow distribution at Arctic sites," *The Cryosphere*, vol. 12, p. 247–270, 2018.

[4] A. M. Stuart, "Inverse problems: A Bayesian perspective," *Acta Numerica*, vol. 19, pp. 451–559, May 2010.

[5] M. A. Iglesias, J. H. Law, and A. M. Stuart, "Ensemble Kalman methods for inverse problems," *Inverse Problems*, vol. 29, no. 4, p. 045001, 2013.

[6] C. Schillings and A. M. Stuart, "Analysis of the Ensemble Kalman Filter for Inverse Problems," *SIAM Journal on Numerical Analysis*, vol. 55, no. 3, pp. 1264–1290, 2017.

[7] M. Iglesias and Y. Yang, "Adaptive regularisation for ensemble kalman inversion," *Inverse Problems*, vol. 37, no. 2, p. 025008, 2021.

[8] D. J. C. MacKay, *Information Theory, Inference, and Learning Algorithms*. Cambridge University Press, 2003.

[9] S. Särkkä, *Bayesian Filtering and Smoothing*. Cambridge University Press, 2013.

[10] E. Jaynes, *Probability Theory: The Logic of Science*. Cambridge University Press, 2003. doi:10.1017/CBO9780511790423.

[11] A. Gelman, J. Carlin, H. Stern, D. Dunson, A. Vehtari, and D. Rubin, *Bayesian Data Analysis*. Chapman and Hall/CRC, 3 ed., 2013.

[12] A. Carrassi, M. Bocquet, L. Bertino, and G. Evensen, "Data assimilation in the geosciences: An overview of methods, issues, and perspectives," *WIREs Climate Change*, vol. 9, Sept. 2018.

[13] G. Evensen, F. C. Vossepoel, and P. J. van Leeuwen, *Data Assimilation Fundamentals*. Springer, 2022.

[14] M. Katzfuss, R. S. Stroud, and C. K. Wikle, "Ensemble Kalman Methods for High-Dimensional Hierarchical Dynamic Space-Time Models," *Journal of the American Statistical Association*, vol. 115, no. 530, pp. 866–885, 2020.

[15] R. M. Neal, "Sampling from multimodal distributions using tempered transitions," *Statistics and Computing*, vol. 6, pp. 353–366, Dec. 1996.

[16] A. S. Stordal and A. H. Elsheikh, "Iterative ensemble smoothers in the annealed importance sampling framework," *Advances in Water Resources*, vol. 86, pp. 231–239, Dec. 2015.

[17] A. Garbuno-Inigo, F. Hoffmann, W. Li, and A. M. Stuart, "Interacting langevin diffusions: Gradient structure and ensemble kalman sampler," *SIAM Journal on Applied Dynamical Systems*, vol. 19, pp. 412–441, Jan. 2020.

[18] E. Cleary, A. Garbuno-Inigo, S. Lan, T. Schneider, and A. M. Stuart, "Calibrate, emulate, sample," *Journal of Computational Physics*, vol. 424, p. 109716, Jan. 2021.

[19] O. Dunbar, A. Duncan, A. Stuart, and M.-T. Wolfram, "Ensemble Inference Methods for Models With Noisy and Expensive Likelihoods," *SIAM Journal on Applied Dynamical Systems*, vol. 21, no. 2, pp. 1539–1572, 2022.

[20] N. Papadakis, E. Mémin, A. Cuzol, and N. Gengembre, "Data assimilation with the weighted ensemble Kalman filter," *Tellus A: Dynamic Meteorology and Oceanography*, vol. 62, pp. 673–697, Jan. 2010.

[21] N. Chopin and O. Papaspiliopoulos, *An Introduction to Sequential Monte Carlo*. Springer, 2020.

[22] G. Bretthorst, *Bayesian Spectrum Analysis and Parameter Estimation*. Springer, 1988.

[23] K. Aalstad, S. Westermann, and L. Bertino, "Evaluating satellite retrieved fractional snow-covered area at a high-Arctic site using terrestrial photography," *Remote Sensing of Environment*, vol. 239, p. 111618, 2020.

[24] L. D. van der Valk, A. J. Teuling, L. Girod, N. Pirk, R. Stoffer, and C. C. van Heerwaarden, "Understanding wind-driven melt of patchy snow cover," *The Cryosphere*, 2022.

[25] P. L. Finkelstein and P. F. Sims, "Sampling error in eddy correlation flux measurements," *Journal of Geophysical Research: Atmospheres*, vol. 106, pp. 3503–3509, Feb. 2001.

[26] E. Bassi, "From Here to 2023: Civil Drones Operations and the Setting of New Legal Rules for the European Single Sky," *Journal of Intelligent & Robotic Systems*, vol. 100, pp. 493–503, Nov. 2020.

[27] R. T. Palomaki, N. T. Rose, M. van den Bossche, T. J. Sherman, and S. F. J. De Wekker, "Wind Estimation in the Lower Atmosphere Using Multirotor Aircraft," *Journal of Atmospheric and Oceanic Technology*, vol. 34, pp. 1183–1191, May 2017.